# SKARL: Provably Scalable Kernel Mean Field Reinforcement Learning for Variable-Size Multi-Agent Systems

## Abstract

Scaling multi-agent reinforcement learning (MARL) requires both scalability to large swarms and flexibility across varying population sizes. A promising approach is mean-field reinforcement learning (MFRL), which approximates agent interactions via population averages to mitigate state-action explosion. However, this approximation has limited representational capacity, restricting its effectiveness in truly large-scale settings. In this work, we introduce Scalable Kernel MeAn-Field Multi-Agent Reinforcement Learning (SKARL), which lifts this bottleneck by embedding agent interactions into a reproducing kernel Hilbert space (RKHS). This kernel mean embedding provides a richer, size-agnostic representation that enables scaling across swarm sizes without retraining or architectural changes. Furthermore, a cylindrical kernel function is introduced to ensure universal approximation over functional space. For efficiency, we design an implementation based on functional gradient updates with Nyström approximations, which makes kernelized mean-field learning computationally tracable. From the theoretical side, we establish convergence guarantees for both the kernel functionals and the overall SKARL algorithm. Empirically, SKARL trained with 64 agents generalizes seamlessly to deployments ranging from 4 to 256 agents, outperforming MARL baselines.

## 1 Introduction

Multi-agent reinforcement learning (MARL) has achieved remarkable progress in domains such as multi-robot coordination (Vinyals et al., 2019; Berner et al., 2019). However, scaling MARL to large populations remains a fundamental challenge (Du et al., 2023). As the number of agents increases, the joint state–action space grows exponentially, and interaction dynamics become increasingly complex. This induces a curse of dimensionality that makes conventional learning unstable and inefficient (Tan, 1993; Tampuu et al., 2015). Moreover, most existing MARL methods lack population scalability: policies trained with one swarm size often fail to generalize to other scales in zero-shot. These limitations naturally raise the question: How can we design MARL algorithms that scale efficiently to hundreds of agents while generalizing seamlessly to unseen population sizes?

A promising direction is the use of mean-field approximations (Caines et al., 2006; Lasry & Lions, 2007). By summarizing agent interactions through a population distribution, mean-field MARL (MFRL) (Yang et al., 2018) avoids exponential complexity growth and exploits the permutation invariance of homogeneous swarms. Prior work has demonstrated the feasibility of mean-field methods in large-scale settings (Angiuli et al., 2021; Gu et al., 2025). However, how to design a universally effective way to represent the population distribution remain a bottleneck. Traditional distribution representation paradigms in the field of mean-field mainly fall into two categories. The first method employs spatial discretization techniques such as histograms (Carmona et al., 2019) and $\epsilon$-net (Gu et al., 2021), which preserve distributional information with theoretical guarantee but suffer from the curse of dimensionality in high-dimensional spaces, as the number of discrete units grows exponentially with state dimensions. The second relies on statistical moments, ranging from first-order means (Yang et al., 2018) to higher-order statistics (Pham & Warin, 2023). This paradigm adapts well to high-dimensional state spaces as moment calculations avoid explicit space partitioning, but the representational capacity is limited, as the conflation of distinct distributions and lack of

critical structural details like multi-modality. As a result, current mean-field approaches still struggle to achieve scalability when applied to sufficiently large populations.

In recent years, kernel-based methods have emerged as the third direction for distribution representation (Wang et al., 2020b; Liu et al., 2020; Cui et al., 2023; Fiedler et al., 2023; 2025), aiming to combine the scalability of moment-based methods with the expressiveness of discretization-based approaches. In the context of mean-field systems, these methods leverage kernel functions (e.g. radial basis function kernel) to embed population distributions into high-dimensional feature spaces, transforming distribution-level interactions into tractable feature operations (Wang et al., 2020b; Liu et al., 2020; Cui et al., 2023). Nevertheless, existing kernel-based methods rely on fixed kernel functions or constrained feature structures, failing to guarantee that their representation space can fully span all possible population distributions, especially when agent number tends to larger. This incompleteness in representational coverage may lead to missed critical distributional characteristics, ultimately restricting the scalibility performance.

To address this problem, we introduce Scalable Kernel MeAn-Field Multi-Agent Reinforcement Learning (SKARL), a novel approach that integrates mean-field learning with reproducing kernel Hilbert space (RKHS) representations to achieve both scalability and flexibility. Unlike traditional kernel methods constrained by fixed structures, SKARL employs kernel mean embeddings to map the entire population distribution into the RKHS, capturing intrinsic structural details (e.g., multi-modality) in a size-agnostic manner. Furthermore, to ensure the global approximation, we model the $Q$-function for individual agent as a cylindrical kernel functional, inspired by Guo et al. (2023), and derive functional gradient updates under a dual time-scale learning scheme. To ensure computational efficiency in large populations, we employ Nyström approximations to project functional updates onto low-dimensional subspaces (Williams & Seeger, 2000). Our framework offers both theoretical and empirical benefits. We prove that cylindrical kernel functionals form a universal approximator over distribution spaces, ensuring expressiveness, and establish that the resulting value functions are Wasserstein-Lipschitz continuous, providing robustness to distributional shifts. Crucially, by representing the swarm as a distribution rather than a fixed-size set, our method naturally supports population flexibility to 4 times larger agent size in deployment compared with training phase. Empirically, SKARL achieves superior performance on large-scale cooperative tasks, consistently outperforming MARL baselines with and without mean-field techniques in cumulative reward and training stability.

In summary, our contributions are as follows:

- We propose the SKARL, a novel MARL framework that integrates RKHS distribution embedding with mean-field multi-agent reinforcement learning, providing a size-agnostic, distribution-level representation beyond moments and fixed kernel embedding representations.

- We model individual Q-functions as cylindrical kernel functionals over the embedded population distribution, significantly enhancing expressive capacity compared with traditional parametric critics.

- We develop a functional gradient algorithm for cylindrical kernel functionals, along with a dual time-scale learning rule and Nyström approximations for efficiency. Theoretically, we prove universal approximation and establish Wasserstein-Lipschitz continuity of the value functions.

- Through extensive experiments on large-scale benchmarks, we demonstrate that SKARL generalizes seamlessly across population sizes and achieves significant improvements over MARL baselines in both performance and stability.

## 2 PRELIMINARIES

### 2.1 MULTI-AGENT STOCHASTIC GAME

We consider an episodic mean-field reinforcement learning game with a fixed number of agents $N \in \mathbb{N}$. Such a game is defined by the tuple $\langle \mathcal{S}^N, \mathcal{A}^N, P, (r^i)_{i=1}^N, \gamma \rangle$, where $\mathcal{S}^N = \mathcal{S}_1 \times \cdots \times \mathcal{S}_N$ denotes the joint state space: a vector $\boldsymbol{s} = (s^1, \ldots, s^N)$ collects the local state $s^i \in \mathcal{S}_i$ of each agent. Similarly, the joint action space is $\mathcal{A}^N = \mathcal{A}_1 \times \cdots \times \mathcal{A}_N$, where a joint action $\boldsymbol{a} = (a^1, \ldots, a^N)$

consists of local actions $a^i \in \mathcal{A}_i$. In the homogeneous setting, agents share the same state and action spaces, i.e., $\mathcal{S} = \mathcal{S}_1 = \cdots = \mathcal{S}_N$ and $\mathcal{A} = \mathcal{A}_1 = \cdots = \mathcal{A}_N$. System dynamics are governed by a stochastic kernel $P : \mathcal{S}^N \times \mathcal{A}^N \to \mathcal{P}(\mathcal{S}^N)$, where $\mathcal{P}(\mathcal{S}^N)$ denotes the set of probability measures over $\mathcal{S}^N$. Each agent receives an instantaneous reward $r^i(s, a) = r(s^i, a^i)$, which couples individual behavior with the global population. Finally, $0 < \gamma < 1$ is the discount factor weighting future returns. The objective is to learn a joint policy $\pi = (\pi^1, \ldots, \pi^N)$, where each $\pi^i : \mathcal{S} \to \mathcal{P}(\mathcal{A})$, that maximizes for every agent $i$ the expected discounted return

$$J^i(\pi) \;=\; \mathbb{E}_{s_0 \sim d_0, \, P, \, \pi}\Big[\sum_{t=0}^{T-1} \gamma^t \, r^i(s_t, a_t)\Big],$$

with the expectation taken over the initial state distribution $d_0$, the transition kernel $P$, and the stochastic choices of the joint policy $\pi$.

## 2.2 MEAN FIELD REINFORCEMENT LEARNING

In multi-agent reinforcement learning with $N$ agents, the Q-function of agent $i$ depends on the joint action $\boldsymbol{a} = (a^1, \ldots, a^N)$, where each $a^j$ is represented by a one-hot vector. This leads to an exponential blow-up of the action space, a manifestation of the curse of dimensionality. Mean-field reinforcement learning (MFRL) (Yang et al., 2018) addresses this by approximating pairwise interactions through a mean-field term. Specifically, the Q-function of agent $i$ is written as

$$Q^i(s, \boldsymbol{a}) = \frac{1}{N_i} \sum_{j \in \mathcal{N}^i} Q^i(s, a^i, a^j) \approx Q^i\left(s, a^i, \bar{a}^{-i}\right),$$

where $\bar{a}^{-i} := \frac{1}{N_i} \sum_{j \in \mathcal{N}^i} a^j$ denotes the empirical mean action of agent $i$'s neighbors $\mathcal{N}^i$ with size $N_i$. This induces a dynamical system in which each agent responds to the mean-field action via a softmax policy as $\pi_t^i(\cdot \mid s) = \mathrm{softmax}\big(-\beta Q_t^i(s, \cdot, \bar{a}_t^{-i})\big)$, where the softmax is taken over all $a \in \mathcal{A}$.

For continuous action spaces, the mean-field action is modeled as a distribution on the 2-Wasserstein space $\mathcal{P}_2(\mathcal{A})$ (Guo & Xu, 2019):

$$\nu^{-i} = \frac{1}{N_i} \sum_{j=1}^{N_i} \delta_{a^j},$$

where $\delta_{a^j}$ is the Dirac measure at action $a^j$. If the pairwise Q-function is twice Lions-differentiable with respect to the mean-field action $\mu_{a_j}$, the Lions–Taylor expansion yields (Tang et al., 2024)

$$Q^i(s, \boldsymbol{a}) \;\approx\; \bar{Q}^i(s, a^i, \mu^{-i}) + \frac{1}{N_i} \sum_{j=1}^{N_i} \partial_\nu \bar{Q}^i(s, a^i, \mu^{-i})[a^j] \cdot (\bar{a}^i - a^j), \tag{1}$$

where $\bar{Q}^i(s, a^i, \delta_{a_j})$ is the Q-function lifted to the Wasserstein space, $\bar{a}^i = \frac{1}{N_i} \sum_j a^j$ is the mean neighbor action, and $\partial_\nu \bar{Q}^i(s, a^i, \mu^{-i})[\cdot] : \mathcal{A} \to \mathcal{A}$ is the Lions derivative such that for any sequence $\{\nu_n\} \subset \mathcal{P}(\mathcal{A})$ with the norm-2 Wasserstein distance converges to 0, when $n \to 0$, it always hold that

$$\frac{Q(s, a^i, \nu_n) - Q(s, a^i, \nu) - \int_{\mathcal{A}^2} \partial_\nu Q(s, a^i, \nu)(x) \cdot (y - x) \pi(\mathrm{d}x, \mathrm{d}y)}{\mathcal{W}_2^2(\nu_n, \nu)} \to 0,$$

where $\pi_n \subset \mathcal{P}_2(\mathcal{A} \times \mathcal{A})$ denotes the optimal plan between $\nu_n$ and $\nu$.

## 2.3 RKHS AND KERNEL MEAN EMBEDDING IN MFRL

In recent years, kernel-based methods have been successfully integrated into MFRL to enhance the expressiveness and scalability of value functions and policies. A key tool for this integration is the RKHS, which provides a nonparametric framework for representing distributions over agent states (). An RKHS ($\mathcal{H}_k$) over the domain ($\mathcal{X}$) is a Hilbert space of functions ($g : \mathcal{X} \to \mathbb{R}$) associated with a symmetric positive-definite kernel ($k : \mathcal{X} \times \mathcal{X} \to \mathbb{R}$). The defining property of an RKHS is the reproducing identity (Muandet et al., 2017):

$$g(x) = \langle g, k(x, \cdot) \rangle_{\mathcal{H}_k} = \int_{\mathcal{X}} g(x') k(x, x'), \mathrm{d}x'. \tag{2}$$

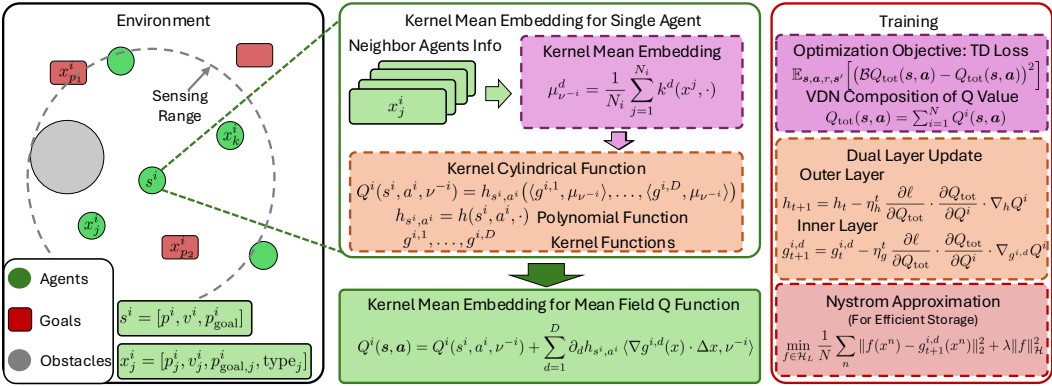

Figure 1: Overview of the SKARL framework. Agent interactions are embedded into RKHS via kernel mean embeddings and evaluated through kernel cylindrical functions to approximate mean-field Q-values. Updates are performed with temporal-difference learning and Nyström projection for scalability and efficiency.

In the context of MFRL, the kernel mean embedding (KME) method is used to represent the distribution of agents in a high-dimensional feature space. For any distribution $\mathbb{P}$ over $(\mathcal{X})$, its embedding is defined as:

$$\mu_{\mathbb{P}}(\cdot) := \mathbb{E}_{X \sim \mathbb{P}}[k(\cdot, X)] = \int_{\mathcal{X}} k(\cdot, x), d\mathbb{P}(x), \tag{3}$$

which is a mapping from the distribution to an element of the RKHS. This embedding preserves expectations, such that for any function $g \in \mathcal{H}_k$, the expected value of $g(X)$ under the distribution $\mathbb{P}$ is given by the inner product in $\mathcal{H}_k$ as $\mathbb{E}_{X \sim \mathbb{P}}[g(X)] = \langle g, \mu_{\mathbb{P}} \rangle_{\mathcal{H}_k}$. This allows the representation of the entire mean-field distribution of agents as a single element in the RKHS, facilitating efficient computation and flexible modeling of the agent population's behavior.

## 3 THE SKARL FRAMEWORK

This section presents the derivation of SKARL within the Reproducing Kernel Hilbert Space (RKHS), as is shown in Figure. 1. In this work, we aim to design a population representation for MFRL that is both expressive and scalable. Classical spatial discretization methods provide theoretical guarantees but suffer from the curse of dimensionality, as the number of cells explodes with the state dimension, making them unsuitable for large-scale RL (Carmona et al., 2019; Gu et al., 2021) Moment-based approaches alleviate this by summarizing populations via first-order means (Yang et al., 2018) or a few higher-order statistics Pham & Warin (2023), yet they fundamentally discard fine-grained distributional structure and struggle to capture complex agent interactions. More recent kernel-based methods strike a middle ground between these two extremes, but typically rely on fixed kernels or rigid feature parameterizations, which do not guarantee that all relevant population distributions can be well approximated (Cui et al., 2023) Motivated by these limitations, we build our framework on KME to represent populations in a high-dimensional RKHS, and further introduce a cylindrical kernel functional that endows the representation with global approximation capabilities over the space of mean-field distributions.

### 3.1 KERNEL MEAN EMBEDDING REPRESENTATION OF MEAN-FIELD Q-FUNCTIONS

**Mean-Field Embedding via KME** The mean-field measure is embedded via empirical KME:

$$\mu_{\nu^{-i}}^d = \frac{1}{N_i} \sum_{j=1}^{N_i} k^d(x^j, \cdot),$$

where $x^j$ is the latent embedding of neighbor $(s^j, a^j)$.

**Kernel Cylindrical Representation of Pairwise Interactions.** Mean-field Q-functions are functionals of probability measures over neighbor actions. To approximate such distributional functionals in a principled and expressive way, we introduce *kernel cylindrical functions*, inspired by work of Guo et al. (2023), which provide universal approximations within RKHS. Formally, for any continuous functional $f : \mathcal{P}(\mathcal{M}) \to \mathbb{R}$ with bounded Lions derivatives, we approximate it by $h(\nu)$ with definition as

$$h(\nu) := h\big(\langle g^1, \mu_\nu^1 \rangle_{\mathcal{H}_k}, \ldots, \langle g^D, \mu_\nu^D \rangle_{\mathcal{H}_k}\big), \tag{4}$$

where each $g^d(\cdot) = k(x^d, \cdot)$ is a kernel anchored at $x^d \in \mathcal{M}$, $\mu_\nu^d$ denotes the empirical KME, and $h : \mathbb{R}^D \to \mathbb{R}$ is a polynomial function with parameters $\theta_h$, with $D$ denotes the number of used kernels and $d$ denotes the index of kernel. The inner products $\langle g^d, \mu_\nu^d \rangle_{\mathcal{H}_k} = \int_{\mathcal{M}} g^d(x) \, d\nu(x)$ serve as kernel-based summaries of $\nu$. Base on this function type, we develop the following theorem, implying that any smooth mean-field Q-function can be approximated arbitrarily well by such cylindrical representations.

**Theorem 3.1** (Density of Kernel Cylindrical Functions). *Let $\mathcal{P}(\mathcal{M})$ be the space of Borel probability measures over a compact manifold $\mathcal{M} \subset \mathbb{R}^d$. Define*

$$\mathcal{G}_D(\mathcal{M}) := \big\{ h(\mu) = h\big(\langle g^1, \mu \rangle_{\mathcal{H}_k}, \ldots, \langle g^D, \mu \rangle_{\mathcal{H}_k}\big) \,\big|\, h \text{ (polynomial)}, \{g^d\}_{d=1}^D \text{ kernels} \big\}. \tag{5}$$

*Let $\mathcal{C}^{1,1}(\mathcal{M})$ denote the space of Fréchet differentiable functions with Lipschitz derivatives. Then, for any $f \in \mathcal{C}^{1,1}(\mathcal{M})$ and any $\epsilon > 0$, there exists $h \in \mathcal{G}_D(\mathcal{M})$ such that $|f(\mu) - h(\mu)| < \epsilon$ for all $\mu \in \mathcal{P}(\mathcal{M})$, provided $D$ is sufficiently large.*

This directly yields a representation of the pairwise interaction in agent $i$'s Q-function:

$$Q^i(s^i, a^i, \nu^{-i}) = h_{s^i, a^i}\big(\langle g^{i,1}, \mu_{\nu^{-i}} \rangle, \ldots, \langle g^{i,D}, \mu_{\nu^{-i}} \rangle\big),$$

where $h_{s^i, a^i}(\cdot) = h(s^i, a^i, \cdot) : \mathbb{R}^D \to \mathbb{R}$ is differentiable with its parameters, and $g^{i,d} = \sum_{m=1}^M \alpha_m^d k^d(x^m, \cdot)$, with anchor points $\{x^m\}_{m=1}^M$ in latent space $\mathcal{X}$ and learnable weights $\{\alpha_m^d\}$. The gradient of $g^{i,d}$ is $\nabla g^{i,d}(x) = \sum_m \alpha_m^d \partial_x k^d(x^m, x)$. To guaranty continuity, we assume Lipschitz continuity and boundedness of kernels.

**Assumption 3.1** (Lipschitz Continuity and Boundedness). *Each kernel $g^{i,d}$ is $L_g$-Lipschitz:*

$$|g^d(x) - g^d(y)| \le L_g \|x - y\|_2, \quad \forall x, y \in \mathcal{X},$$

*and uniformly bounded: $|k(x, y)| < \infty, \quad \forall x, y \in \mathcal{X}$. Without loss of generality, assume $\sup_{x \in \mathcal{X}} |k(x, x)| \le 1$.*

The Lions derivative of a cylindrical function $h(\nu)$ is (Guo et al., 2023):

$$\partial_\nu h(\nu)(x) = \sum_{d=1}^D \partial_d h(\nu) \, \nabla g^d(x),$$

where $\partial_d h$ denotes the derivative with respect to the $d$-th argument.

**Local Value Function Approximation.** Combining state-action embeddings, cylindrical functionals, and mean-field embeddings yields a computational representation of the local Q-function. Analogous to Eq. (1), we approximate

$$Q^i(\boldsymbol{s}, \boldsymbol{a}) = h_{s^i, a^i}\big(\langle g^{i,1}, \mu_{\nu^{-i}} \rangle, \ldots, \langle g^{i,D}, \mu_{\nu^{-i}} \rangle\big) + \sum_{d=1}^D \partial_d h_{s^i, a^i} \, \langle \nabla g^{i,d}(x) \cdot \Delta x, \nu^{-i} \rangle, \tag{6}$$

where $\Delta x := \bar{x}^i - x$ and $\bar{x}^i = \frac{1}{N_i} \sum_j x^j$. The first term captures mean-field interactions, while the second encodes gradient corrections.

This representation integrates seamlessly with standard multi-agent value-decomposition methods such as VDN (Sunehag et al., 2017), QMIX (Rashid et al., 2018), and QPLEX (Wang et al., 2020a). Analogous constructions apply to the state-value function $V^i(\boldsymbol{s})$ and advantage function $A^i(\boldsymbol{s}, \boldsymbol{a})$.

## 3.2 VALUE FUNCTION UPDATE WITH STORAGE EFFICIENCY

**Updating Cylindrical Kernel Functions.** The total value function $Q_{\text{tot}}$ is decomposed into agent-wise functions $Q^i$ under the Individual Global Max (IGM) principle (Rashid et al., 2018) (See Appendix E). To update $Q^i$, we optimize the temporal-difference (TD) loss (Sutton, 1988)

$$\ell(\mathcal{B}Q_{\text{tot}}, Q_{\text{tot}}) = \mathbb{E}_{\boldsymbol{s},\boldsymbol{a},r,\boldsymbol{s}'}\left[\left(\mathcal{B}Q_{\text{tot}}(\boldsymbol{s},\boldsymbol{a}) - Q_{\text{tot}}(\boldsymbol{s},\boldsymbol{a})\right)^2\right],$$

where $\mathcal{B}$ denotes the Bellman operator (Puterman, 1994), i.e.,

$$(\mathcal{B}Q_{\text{tot}})(\boldsymbol{s},\boldsymbol{a}) = \mathbb{E}_{\boldsymbol{s}'}\left[r(\boldsymbol{s},\boldsymbol{a}) + \gamma \max_{\boldsymbol{a}'} Q_{\text{tot}}(\boldsymbol{s}',\boldsymbol{a}')\right].$$

Parameters are updated by gradient descent in two spaces. For the outer function $h$ and RKHS components $\{g^{i,d}\}$, with learning rate $\eta_h^t, \eta_g^t$.

$$\theta_h^{t+1} = \theta_h^t - \eta_h^t \frac{\partial \ell}{\partial Q_{\text{tot}}} \cdot \frac{\partial Q_{\text{tot}}}{\partial Q^i} \cdot \nabla_h Q^i, \; g_{t+1}^{i,d} = g_t^{i,d} - \eta_g^t \frac{\partial \ell}{\partial Q_{\text{tot}}} \cdot \frac{\partial Q_{\text{tot}}}{\partial Q^i} \cdot \nabla_{g^{i,d}} Q^i, \qquad (7)$$

where $\{g^{i,d}\}$ are updated via the Fréchet derivative.

**Proposition 3.2** (Fréchet Derivative Form)**.** *The Fréchet derivative of $Q^i$ with respect to $g^{i,d}$ decomposes as*

$$\nabla_{g^{i,d}} Q^i = \underbrace{\left(\partial_d h + \sum_{d'} \partial_{dd'}^2 h \left\langle \nabla g^{d'} \cdot \Delta x, \nu^{-i} \right\rangle\right) \mu_{\nu^{-i}}}_{\textit{Mean interaction term}} - \underbrace{\partial_d h \nabla \cdot (\nu^{-i} \Delta x)}_{\textit{Divergence term}}, \qquad (8)$$

*where $\Delta x := \bar{x}^i - x$. See Remark D.3 in the Appendix for the explicit form with $N_i$ neighbors.*

**Nyström Approximation for Efficient Storage.** The direct updates in Eq. (7) face two key challenges: (i) the divergence term lies outside the RKHS (Remark D.3), and (ii) naive implementation requires storing $O(N_i T)$ kernels per agent after $T$ iterations, which is infeasible for large swarms and long horizons. To address this, we apply the Nyström approximation, projecting updated functions onto a low-dimensional kernel subspace. Let the anchor set for $g_{t+1}^{i,d}$ be $\{x^n\}_{n=1}^{N_i+M} := \{x^j\}_{j=1}^{N_i} \cup \{x^m\}_{m=1}^{M}$, where $\{x^m\}$ are anchor points from $g_t^{i,d}$ and $\{x^j\}$ are inputs from $\nu$. We select a subset of landmark points $\{z^l\}_{l=1}^{L} \subset \{x^n\}$, spanning an $L$-dimensional subspace $\mathcal{H}_L \subset \mathcal{H}$. The projection of $g_{t+1}^{i,d}$ onto $\mathcal{H}_L$ via Tikhonov regularization is:

$$\tilde{g}_{t+1}^{i,d} = \arg \min_{f \in \mathcal{H}_L} \frac{1}{N_i + M} \sum_{n=1}^{N_i+M} \|f(x^n) - g_{t+1}^{i,d}(x^n)\|_2^2 + \lambda \|f\|_{\mathcal{H}}^2. \qquad (9)$$

By the representer theorem (Schölkopf & Smola, 2002), the solution takes the form $\tilde{g}_{t+1}^{i,d} = \sum_{l=1}^{L} \alpha_l^d k^d(z^l, \cdot)$. Let $\boldsymbol{K}_{LL}^d := [k^d(z^l, z^{l'})]_{1 \leq l,l' \leq L}$ and $\boldsymbol{K}_{N_i+M,L}^d := [k^d(x^n, z^l)]_{1 \leq n \leq N_i+M, 1 \leq l \leq L}$. Then coefficients $\boldsymbol{\alpha}^d = [\alpha_1^d, \ldots, \alpha_L^d]^\top$ admit the closed-form solution (Rudi et al., 2015):

$$\boldsymbol{\alpha}^d = \left(\boldsymbol{K}_{N_i+M,L}^\top \boldsymbol{K}_{N_i+M,L} + \lambda(N_i+M)\boldsymbol{K}_{LL}^d\right)^\dagger \boldsymbol{K}_{N_i+M,L}^\top \mathbf{b},$$

where $\mathbf{b} \in \mathbb{R}^{N_i+M}$ with entries $\mathbf{b}_n = \langle k(x^n, \cdot), g_{t+1}^{i,d} \rangle_{\mathcal{H}_k}$. Here $\dagger$ denotes the Moore–Penrose pseudoinverse. This reduces kernel storage from $O(N_i T)$ to $O(L)$ with $L \ll N_i T$. In our experiments we use uniform sampling for landmark points $\{z^l\}$; other selection strategies are discussed in Remark D.5.

## 3.3 PROPOSED ALGORITHM

With the components mentioned above, the final proposed algorithm is summarized in Algorithm 1.

---

**Algorithm 1** Mean-Field Cylindrical Kernel Method

---

**Input:** Agent swarm size $N$, number of iterations $M$, trajectory batch size $B$, anchor points number $L$, learning rate $(\eta_h, \eta_g)$

1: Initialize local $Q$ function $Q^i$ with kernel functions $\{g^{i,d}\}_{d=1}^D \leftarrow 0$ and outer function $h^i$ for each agent; initialize trajectory set $\mathcal{T}$.
2: **for** $m = 1, \ldots, M$ **do**
3:    **while** Sampling phase **do**
4:       Sample trajectories using the current policy $\{\pi^i\}_{i=1}^N$ with environment, store in $\mathcal{T}$.
5:    **end while**
6:    Sample $B$ trajectories from $\mathcal{T}$ with length $T$ for each trajectory.
7:    Update the outer function $h$ and $\{g_t^{i,d}\}$ with Eq. (7).
8:    Select new anchor points $\{x_l\}_{l=1}^L$ via methods in Remarks D.5.
9:    Projection updated $\{g_t^{i,d}\}$ to $\{\tilde{g}_t^{i,d}\}$ via Eq. (9) and update $Q^i$ with $\{\tilde{g}_t^{i,d}\}$.
10: **end for**
11: **return** final local $Q$ function $Q^i$.

---

# 4 ANALYSIS OF PROPOSED SKARL

## 4.1 COMPUTATIONAL COMPLEXITY, SCALABILITY, AND FLEXIBILITY

We compare the computational complexity of SKARL with value decomposition methods (e.g., QMIX (Rashid et al., 2018)) and mean-field reinforcement learning (MFRL) (Yang et al., 2018)). Table 1 summarizes the results.

Table 1: Comparison of computational complexity and key metrics. $B$: batch size; $N$: number of agents; $L$: landmark points; $D$: number of kernel features.

| Metric | SKARL | QMIX | MFRL |
|---|---|---|---|
| **Q Function Input Size** | $O(\|\mathcal{S}\| + \|\mathcal{A}\| + D)$ | $O(N\|\mathcal{S}\| + N\|\mathcal{A}\|)$ | $O(\|\mathcal{S}\| + \|\mathcal{A}\|)$ |
| **Computation Complexity** | $O(B(L^2N + L^3)D)$ | $O(BN^2)$ | $O(B)$ |
| **Memory Usage** | $O(DL)$ | $O(N)$ | $O(1)$ |
| **Scalability in $N$** | Linear | Exponential | Linear |

**Q Function Input size.** SKARL avoids the $N\|\mathcal{A}\|$ blow-up in QMIX by using kernel-based embeddings (Eq. 6), with $L \ll N$ and $D \ll N$. MFRL is even simpler, but lacks multi-scale coordination.

**Computation.** Complexity is dominated by kernel projections (Eq. 9), scaling with $B$, $N$, and $L$. QMIX suffers $O(N^2)$ due to its mixing network, while MFRL requires only $O(1)$ per agent. When $L$ grows with $N$ (e.g., $L \approx \sqrt{N}$), SKARL's complexity approaches QMIX—this is the main computational drawback.

**Scalability.** SKARL maintains linear dependence on $N$, unlike QMIX's exponential scaling.

**Flexibility.** SKARL generalizes across swarm sizes. If trained with $N$ agents and deployed with $M$, the approximation error is bounded by $O(N^{-1/d} + M^{-1/d})$, where $d$ is the dimension of the state-action space.

**Theorem 4.1** (Flexibility of Kernel Cylindrical Functions). *Let $\nu_N, \nu_M$ denote the empirical mean-field distributions of swarms with $N$ and $M$ agents, sampled from the same distribution $\nu$. Under Assumption 3.1, for a cylindrical function $h$ there exist constants $C_1, C_2 > 0$ such that*

$$\mathbb{E}\big[|h(\nu_N) - h(\nu_M)|\big] \ \leq \ C_1 N^{-1/d} + C_2 M^{-1/d}.$$

## 4.2 CONVERGENCE AND SUBOPTIMALITY

**Convergence of Cylindrical Functions.** The density result in Theorem 3.1 implies approximation power. We now establish convergence rate with respect to the kernel number $D$.

**Theorem 4.2** (Convergence Rate). *Under Assumption 3.1, let $\tilde{f}(\mu_\nu) = f(\nu)$ be a functional depending on the KME $\mu_\nu$ (Eq. 3). Then with probability at least $1 - \delta$,*

$$|h(\nu) - f(\nu)| \leq \sup_\nu \left\| \frac{\delta \tilde{f}}{\delta \mu_\nu} \right\| \left( \sqrt{\frac{1}{D}} + \sqrt{\frac{2 \log(1/\delta)}{D}} \right).$$

*Thus $h$ converges to $f$ at rate $O(D^{-1/2})$.*

**Convergence of Updates.** For the update rules in Eq. equation 7, convergence follows under Robbins–Monro step-size conditions and two-time-scale separation (Borkar, 2008).

**Assumption 4.1** (Robbins–Monro). *Step sizes $\eta_h$ and $\eta_g$ satisfy $\sum_t \eta = \infty$, $\sum_t \eta^2 < \infty$, and $\lim_{t\to\infty} \eta_g/\eta_h = 0$.*

**Theorem 4.3** (Convergence). *Under Assumptions 3.1 and 4.1, the updates converge to $(h^*, \{g^{i,d,*}\})$ minimizing the Bellman TD loss.*

### 4.3 Error of Nyström Approximation

Although the Nyström method substantially reduces storage and computational cost, this method inevitably introduces approximation error. To ensure the reliability of SKARL, it is therefore essential to quantify error of Nyström approximation. We measure the error of projection as

$$\mathcal{E}(f) = \| f - g_{t+1}^{i,d} \|_{L^2},$$

for $f \in \mathcal{H}$, where $L_k f(x) = \langle f, k(x, \cdot) \rangle_{\mathcal{H}_k}$ is the kernel integral operator (Eq. 2). Intuitively, $\mathcal{E}(f)$ captures the deviation between the projected function and the ideal update.

To analyze this error, we introduce two standard conditions from statistical learning theory:

**Assumption 4.2.** *Define the effective dimension $\mathcal{N}(\lambda) = tr((\lambda I + L_k)^{-1} L_k)$. Assume there exists a constant $C_0 > 0$ independent of $\lambda$ such that for any $\lambda > 0$, $\mathcal{N}(\lambda) \leq C_0 \lambda^{-\gamma}$, for some $0 < \gamma \leq 1$.*

**Assumption 4.3.** *There exists $s \geq 0$, $1 \leq R < \infty$, such that $\| L_k^{-s} f_{\mathcal{H}} \|_{\mathcal{H}} < R$, where $f_{\mathcal{H}} := \arg\min_f \mathcal{E}(f)$.*

Combining the Lipschitz continuity of kernel cylindrical functions (Assumption 3.1) with the above spectral assumptions, we obtain the following finite-sample error bound.

**Theorem 4.4** (Nyström Error Bound). *Under Assumptions 3.1, 4.2, and 4.3, let $\delta \in (0, 1)$ and sufficiently large $N_i + M$. With probability at least $1 - \delta$, the excess error of the Nyström approximation satisfies*

$$\mathcal{E}(\tilde{g}_{t+1}^{i,d}) - \min_{f \in \mathcal{H}} \mathcal{E}(f) \leq C_{k,\gamma} \left( \log \frac{6}{\delta} \right)^2 (N_i + M)^{-\frac{2v+1}{2v+\gamma+1}},$$

*where $v = \min(s, 1/2)$, $\lambda = \|L_k\|(N_i + M)^{-\frac{1}{2v+\gamma+1}}$, and $L \geq C_\lambda \log \frac{12}{\lambda \delta}$. Constants $C_{k,\gamma}, C_\lambda$ depend only on the kernel family.*

Theorem 4.4 shows that the Nyström approximation converges to the optimal RKHS projection at a rate depending on both the eigenvalue decay $\gamma$ and the smoothness parameter $s$. In practice, this means that as the number of anchor points $(N_i + M)$ grows, the approximation error shrinks polynomially fast, and only a logarithmic number of landmark points $L$ (relative to the effective dimension) is needed to achieve near-optimal accuracy. This justifies the use of Nyström projection in SKARL.

## 5 Experiments and Results

### 5.1 Experimental Setup

We evaluate our method following the work of Nayak et al. (2023), with four environments: (i) **Move**: Each agent tries to move as fast as possible and avoid collisions. (ii) **Target**: Each agent tries to reach the assigned goal and avoid collisions. (iii) **Coverage**: Each agent tries to go to a goal and

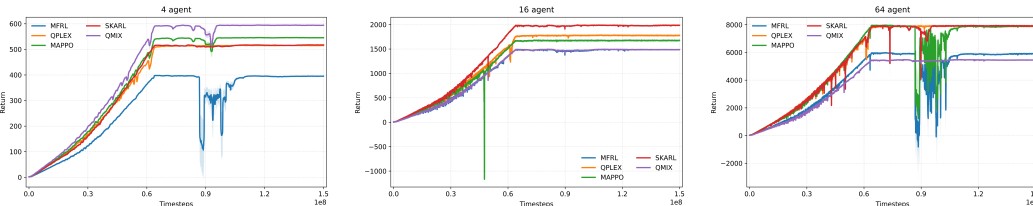

Figure 2: Training results of SKARL and baselines across three environments (5 random seeds).

avoid collisions, and ensure no more than one agent reaching the same goal. (iv) **Line**: There are two landmarks, and the agents try to position themselves equally spread out in a line between the two. For detailed observation, reward and action design, please refer to the Appendix Environments. We compare SKARL against several standard MARL algorithms: (i) **QMIX** (Rashid et al., 2018), (ii) **QPLEX** (Wang et al., 2020a), (iii) **MAPPO** (Yu et al., 2022), (iv) **MFRL** (Yang et al., 2018), and **Fixed-Kernel MFRL** (Cui et al., 2023). For detailed implementation of SKARL and baselines, please refer to Appendix E. We report the test results with 100 max steps.[1].

Table 2: Performance Comparison between SKARL and Baselines in Move Environment

| Algorithm | $N = 4$ | | | $N = 16$ | | | $N = 64$ | | |
|---|---|---|---|---|---|---|---|---|---|
| | R($\uparrow$) | # col($\downarrow$) | S($\uparrow$) | R($\uparrow$) | # col($\downarrow$) | S($\uparrow$) | R($\uparrow$) | # col($\downarrow$) | S($\uparrow$) |
| MAPPO | **947.6** | $0.56_{\pm 0.174}$ | **$0.16_{\pm 0.00779}$** | 3360.2 | $2.6_{\pm 1.12}$ | $0.14_{\pm 0.0562}$ | 14284.8 | $9.6_{\pm 6.98}$ | **$0.15_{\pm 0.0459}$** |
| QMIX | 835.4 | $4.94_{\pm 4.94}$ | $0.15_{\pm 0.0431}$ | 2845.4 | $21.9_{\pm 12.7}$ | $0.13_{\pm 0.08517}$ | 10446.2 | $2.8_{\pm 1.00}$ | $0.11_{\pm 0.0732}$ |
| QPLEX | 911.4 | $0.56_{\pm 0.194}$ | $0.14_{\pm 0.0213}$ | 3625.8 | $20.5_{\pm 10.2}$ | $0.17_{\pm 0.0622}$ | 14073.8 | $22.5_{\pm 9.55}$ | $0.15_{\pm 0.0404}$ |
| MFRL | 734.6 | **$0_{\pm 0}$** | $0.12_{\pm 0.0247}$ | 3083.69 | $38.4_{\pm 6.99}$ | $0.12_{\pm 0.0440}$ | 11411.1 | $204.2_{\pm 22.8}$ | $0.14_{\pm 0.0451}$ |
| Fixed-Kernel | 838.3 | **$0_{\pm 0}$** | $0.14_{\pm 0.0143}$ | 3578.13 | $26.5_{\pm 3.52}$ | $0.15_{\pm 0.0172}$ | 13021.5 | $23.1_{\pm 10.8}$ | $0.15_{\pm 0.0722}$ |
| SKARL | 902.8 | **$0_{\pm 0}$** | $0.15_{\pm 0.0192}$ | **3755.9** | **$12.32_{\pm 5.847}$** | **$0.17_{\pm 0.0500}$** | **14423.8** | $7.9_{\pm 5.37}$ | **$0.15_{\pm 0.0334}$** |

Table 3: Performance Comparison between SKARL and Baselines in Target Environment

| Algorithm | $N = 4$ | | | | $N = 16$ | | | | $N = 64$ | | | |
|---|---|---|---|---|---|---|---|---|---|---|---|---|
| | R($\uparrow$) | T($\downarrow$) | # col($\downarrow$) | S%($\uparrow$) | R($\uparrow$) | T($\downarrow$) | # col($\downarrow$) | S%($\uparrow$) | R($\uparrow$) | T($\downarrow$) | # col($\downarrow$) | S%($\uparrow$) |
| MAPPO | 327.3 | **0.14** | $2.6_{\pm 1.45}$ | **100** | **12** | **0.56** | $8.93_{\pm 5.87}$ | 40.6 | 0 | 1.00 | **0** | 0 |
| QPLEX | 330.3 | 0.18 | $1.3_{\pm 0.982}$ | **100** | -3.1e4 | 1.00 | **$4.92_{\pm 2.55}$** | 0 | -1.9e5 | 1.00 | $19.8_{\pm 12.42}$ | 0 |
| QMIX | 337.0 | **0.14** | **$0.67_{\pm 0.35}$** | **100** | -5.1e4 | 1.00 | $6.6_{\pm 4.28}$ | 0 | -6.4e5 | 1.00 | $41.6_{\pm 18.1}$ | 0 |
| MFRL | 330.8 | **0.14** | $6.4_{\pm 3.38}$ | **100** | 1.1 | 0.81 | $14.375_{\pm 10.28}$ | 31.2 | -5.7e5 | 1.00 | $35.8_{\pm 15.1}$ | 0 |
| Fixed-Kernel | **350.8** | **0.14** | $5.6_{\pm 2.54}$ | **100** | 4.2 | 0.18 | $11.23_{\pm 8.12}$ | 80.2 | 10.2 | **0.97** | $26.7_{\pm 8.2}$ | 1.2 |
| SKARL | 329.3 | 0.18 | $7.2_{\pm 3.15}$ | **100** | 5.6 | 0.96 | $23.2_{\pm 20.5}$ | **100** | **44.75** | 0.98 | $44.3_{\pm 10.6}$ | **3.1** |

## 5.2 MAIN RESULTS

We report the experiment of main experiments on Move and Target environment with 5 random seeds. For other experiments and ablation study, please refer to Appendix F.

**Scale up to large-scale swarms** Figure 2, Table 2 and Table 3 demonstrates SKARL's effectiveness across swarm sizes $N = 4, 16, 64$. We select three metrics: (i) R: global reward. (ii) # col: total collisions. (iii) S: average speed of each agent. For small swarms, SKARL achieves near-optimal reward while entirely eliminating collisions. As the swarm scales to large scale, SKARL outperforms all baselines, achieving the highest reward and fastest speed, with low reduction rate of collision. Notably, SKARL balances safety and efficiency, collisions decrease without sacrificing speed, matching top baselines. These results highlight SKARL's scalability, particularly excelling in mid-to-large swarms where coordination complexity increases.

**Generalize to different swarm sizes** Table 4 and Table 5 reveals SKARL's zero-shot flexibility when tested on varying swarm sizes $M$. When trained on small swarm size, SKARL fails to maintain reasonable performance up to $M = 256$. However, training on larger swarms ($N = 16/64$) enables robust generalization. Most impressively, $N = 64$-trained SKARL achieves near-optimal reward

---

[1]Code at https://anonymous.4open.science/r/SKARL-050D, based on JaxMARL (Rutherford et al., 2023)

Table 4: Zero-Shot Flexibility Performance of SKARL in Move Environment

| Training | Metric | $M=4$ | $M=8$ | $M=16$ | $M=32$ | $M=64$ | $M=128$ | $M=256$ |
|---|---|---|---|---|---|---|---|---|
| $N=4$ | R/N | 225.7 | 168.5 | 177.8 | 155.2 | 166.9 | 168.5 | 173.9 |
| | (# col)/N | $0_{\pm 0}$ | $2.22_{\pm 1.18}$ | $1.36_{\pm 0.794}$ | $0.62_{\pm 0.419}$ | $0.25_{\pm 0.146}$ | $0.12_{\pm 0.0745}$ | $0.22_{\pm 0.0762}$ |
| | S | $0.15_{\pm 0.0190}$ | $0.12_{\pm 0.0842}$ | $0.13_{\pm 0.085}$ | $0.12_{\pm 0.065}$ | $0.11_{\pm 0.0657}$ | $0.11_{\pm 0.0698}$ | $0.12_{\pm 0.0680}$ |
| $N=16$ | R/N | 236.9 | 235.2 | 234.7 | 235.2 | 225.4 | 205.6 | 202.6 |
| | (# col)/N | $0_{\pm 0}$ | $0.98_{\pm 0.437}$ | $0.77_{\pm 0.365}$ | $0.57_{\pm 0.207}$ | $0.17_{\pm 0.115}$ | $0.12_{\pm 0.0652}$ | $0.04_{\pm 0.0221}$ |
| | S | $0.16_{\pm 0.00434}$ | $0.17_{\pm 0.0612}$ | $0.17_{\pm 0.0500}$ | $0.17_{\pm 0.0469}$ | $0.15_{\pm 0.0323}$ | $0.14_{\pm 0.0521}$ | $0.14_{\pm 0.0542}$ |
| $N=64$ | R/N | 231.5 | 221.3 | 227.3 | 224 | 223.2 | 221.6 | 218.7 |
| | (# col)/N | $0_{\pm 0}$ | $0.45_{\pm 0.408}$ | $0.28_{\pm 0.257}$ | $0.44_{\pm 0.275}$ | $0.15_{\pm 0.109}$ | $0.11_{\pm 0.0866}$ | $0.09_{\pm 0.0476}$ |
| | S | $0.15_{\pm 0.0126}$ | $0.15_{\pm 0.0406}$ | $0.15_{\pm 0.0237}$ | $0.16_{\pm 0.0591}$ | $0.15_{\pm 0.0459}$ | $0.15_{\pm 0.0401}$ | $0.15_{\pm 0.0436}$ |
| $N=256$ | R/N | 279.1 | 278.2 | 263.2 | 261.5 | 252.4 | 237.4 | 220.8 |
| | (# col)/N | $0_{\pm 0}$ | $0_{\pm 0}$ | $0.14_{\pm 0.235}$ | $0.28_{\pm 0.254}$ | $0.17_{\pm 0.315}$ | $0.13_{\pm 0.312}$ | $0.10_{\pm 0.451}$ |
| | S | $0.17_{\pm 0.0723}$ | $0.17_{\pm 0.109}$ | $0.16_{\pm 0.124}$ | $0.16_{\pm 0.273}$ | $0.16_{\pm 0.301}$ | $0.16_{\pm 0.334}$ | $0.15_{\pm 0.356}$ |

Table 5: Flexibility Performance of SKARL in Target Environment

| Training | Metric | $M=4$ | $M=8$ | $M=16$ | $M=32$ | $M=64$ | $M=128$ | $M=256$ |
|---|---|---|---|---|---|---|---|---|
| $N=4$ | R/N | 82.3 | -36.25 | -444.0 | -2.8e3 | -9.0e3 | -1.8e4 | -3.7e4 |
| | T (step) | 18 | 95 | 96.5 | 100 | 100 | 100 | 100 |
| | (# col)/N | $0.5_{\pm 0.42}$ | $23_{\pm 14.0}$ | $37.6_{\pm 26.9}$ | $34.875_{\pm 11.34}$ | $46_{\pm 15.1}$ | $138_{\pm 18.9}$ | $342_{\pm 32.8}$ |
| | S% | 100 | 37.5 | 6.25 | 0 | 0 | 0 | 0 |
| $N=16$ | R/N | 85.3 | 7.5 | 0.35 | -2.4e3 | -8.3e3 | -1.7e3 | -2.6e4 |
| | T (step) | 17.4 | 13.8 | 96.3 | 98.5 | 99.4 | 100 | 100 |
| | (# col)/N | $0.4_{\pm 0.13}$ | $19.25_{\pm 13.0}$ | $23.2_{\pm 20.5}$ | $34.875_{\pm 11.34}$ | $46_{\pm 15.1}$ | $75.5_{\pm 14.6}$ | $116_{\pm 21.1}$ |
| | S% | 100 | 100 | 100 | 75 | 6.25 | 0 | 0 |
| $N=64$ | R/N | 84.0 | 77.3 | 69.8 | 10.8 | 0.70 | -0.25 | -10.5 |
| | T (step) | 18.7 | 27.8 | 30.6 | 67.2 | 98.1 | 100 | 100 |
| | (# col)/N | $0.5_{\pm 0.342}$ | $3_{\pm 2.35}$ | $6.7_{\pm 6.45}$ | $16.1_{\pm 5.83}$ | $44.3_{\pm 10.6}$ | $66.3_{\pm 15.2}$ | $96.8_{\pm 17}$ |
| | S% | 100 | 100 | 93.75 | 75 | 12.5 | 0 | 0 |
| $N=256$ | R/N | 87.2 | 80.5 | 78.6 | 22.9 | 20.8 | 15.4 | 12.1 |
| | T (step) | 18.6 | 22.2 | 28.5 | 30.4 | 32.8 | 45.1 | 60.2 |
| | (# col)/N | $0_{\pm 0}$ | $0.5_{\pm 0.412}$ | $1.4_{\pm 0.24}$ | $5.7_{\pm 1.32}$ | $11.2_{\pm 5.2}$ | $12.1_{\pm 9.4}$ | $14.2_{\pm 10.5}$ |
| | S% | 100 | 100 | 100 | 100 | 100 | 98.71 | 95.21 |

per agent at $M = 256$, while collisions remain the lowest. This flexibility stems from SKARL's distribution-driven policy as is in Theorem 4.1, enabling deployment in real-world scenarios where swarm sizes are dynamic.

## 6 CONCLUSION

We propose **SKARL**, a scalable framework for large-scale multi-agent reinforcement learning. **SKARL** resolves the scalability and flexibility bottlenecks of multi-agent reinforcement learning by enabling linear complexity in swarm size and zero-shot transfer across populations. It ensures convergence with efficient updates and drastically reduces training overhead, allowing effective learning in large swarms. Experiments confirm that SKARL outperforms state-of-the-art baselines in both performance and generalization. While our methods offers valuable insights into the representation of mean-field, there are several limitations to consider. Our methods relies on the homogeneous assumption, which limits the application to heterogeneous groups. In the future, we aim to improve the design and extend to heterogeneous MARL problems.

## 7 ETHICS STATEMENT

This work introduces SKARL, a scalable kernel mean-field reinforcement learning framework for large-scale multi-agent systems. Our contributions are primarily theoretical and methodological, with empirical validation performed in simulated multi-agent environments such as swarm naviga-

tion, coordination, and collision avoidance benchmarks. These environments are widely used in the MARL community and do not involve human subjects, sensitive personal data, or proprietary datasets.

We acknowledge that advances in multi-agent reinforcement learning (MARL) may have dual-use implications. While our experiments are limited to academic and open-source benchmarks, similar techniques could be applied in high-stakes domains such as autonomous vehicle fleets, aerial drone swarms, or defense systems. In such settings, ethical concerns may include safety, accountability, and fairness. To mitigate potential risks, our work remains focused on theoretical scalability and generalization, and we refrain from proposing or testing direct real-world deployment scenarios.

From a fairness perspective, the algorithms studied here are agnostic to sensitive human attributes and do not incorporate demographic information. From a privacy and security perspective, no personal or confidential information is processed. From a research integrity perspective, we strictly adhere to reproducible and transparent reporting, with proofs, assumptions, and algorithms explicitly documented. Finally, we affirm that we have read and adhered to the ICLR Code of Ethics, and have conducted this research in alignment with its principles.

## 8 REPRODUCIBILITY STATEMENT

We have undertaken comprehensive steps to ensure that the theoretical and empirical results reported in this paper are reproducible. For the theoretical contributions, all assumptions are explicitly stated, and full mathematical proofs are provided either in the main text or in the appendix. These proofs establish the universal approximation property of kernel cylindrical functions and the convergence of the dual time-scale learning rule.

For the empirical results, all experiments are conducted on widely used benchmark environments for multi-agent reinforcement learning, such as large-scale swarm coordination tasks. We describe the experimental setup, training protocols, and hyperparameter configurations in detail within the paper and provide additional clarifications in the appendix. Random seeds are fixed across runs, and ablation studies are reported to verify stability.

To further facilitate reproducibility, we release anonymous source code, including implementations of SKARL, training scripts, and environment configuration files, as part of the supplementary materials. This enables other researchers to directly reproduce the results presented in this paper, adapt the framework to new environments, or verify the theoretical guarantees with empirical evidence. Together, these measures ensure that the community can reliably replicate and build upon our contributions.

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

## A  THE USE OF LLM

In the preparation of this paper, we employed large language models (LLMs) strictly as assistive tools. Their role was confined to three aspects: (i) improving the clarity and readability of the manuscript by suggesting stylistic refinements and alternative phrasings; (ii) assisting with the organization and presentation of mathematical proofs, including the checking of algebraic manipulations and the polishing of logical exposition; and (iii) serving as a coding assistant for routine programming tasks such as code completion, debugging, and documentation generation.

Importantly, LLMs were not involved in the generation of research ideas, the design of the SKARL framework, or the conceptual development of the theoretical results. All scientific insights, algorithmic designs, and experimental implementations originate from the authors. The LLM usage did not extend to generating novel theorems, creating data, or drawing conclusions. Instead, the models functioned in a supportive capacity, comparable to grammar-checking or code editor auto-completion, with the final responsibility for correctness, originality, and integrity resting solely on the authors.

We disclose this usage in alignment with ICLR policy. By transparently reporting the scope of assistance, we affirm that the LLMs were used responsibly and ethically, and that the intellectual contributions of this work are entirely attributable to the authors.

## B  PROOFS OF THEOREMS, LEMMAS AND PROPOSITIONS

### B.1  PROOF OF THEOREM 3.1

*Proof.* We establish the density of proposed $\mathcal{G}_D(\mathcal{M})$. To this end, we first need:

**Lemma B.1** (Stone–Weierstrass)**.** *Take a compact Hausdorff space $H$, and let $\mathcal{C}(H)$ be the algebra of real-valued continuous functions on $H$, with the topology of uniform convergence. Let $\mathcal{A}$ be a subalgebra of $\mathcal{C}(H)$. If $\mathcal{A}$ separates points on $H$ and vanishes at no point on $H$, then $\mathcal{A}$ is dense in $\mathcal{C}(H)$.*

Then, following the proof of Lemma 3.12 in Guo et al. (2023), we prove that with appropriate choices of norms, $\mathcal{G}_D(\mathcal{M})$ is dense in $\mathcal{C}^{1,1}(\mathcal{M})$.

**Lemma B.2.** $\mathcal{G}_D(\mathcal{M})$ *is dense in $\mathcal{C}^{1,1}(\mathcal{M})$ with the supremum norm of derivatives of all orders: for $\Phi \in \mathcal{C}^{1,1}(\mathcal{M})$,*

$$\|\Phi\|_{\mathcal{M}} := \sup_{(\nu,x)\in\mathcal{P}(\mathcal{M})\times\mathcal{M}} \Big( |\Phi(\nu)| + |\partial_\mu \Phi(\mu)(x)| + \|\partial_x \partial_\mu \Phi(\mu)(x)\| \Big)$$

We prove this with two steps:

**Step 1**: take $\Phi \in \mathcal{C}^{1,1}(\mathcal{M})$, then $\partial_{xx}\frac{\delta\Phi}{\delta\mu}(\mu,x)$ is a continuous function on $\mathcal{P}(\mathcal{M})\times\mathcal{M}$ by definition, namely, $\partial_{xx}\frac{\delta\Phi}{\delta\mu}(\mu,x) \in \mathcal{C}(\mathcal{P}(\mathcal{M})\times\mathcal{M})$. Define the algebraic space that contains $\mathcal{G}_D(\mathcal{M})$ for some $n \in]mathbbN$ as

$$\mathcal{H}(\mathcal{P}(\mathcal{M})\times\mathcal{M}) := \Big\{ \Phi(\mu,x) = \sum_{k=1}^{n} f^k(\langle g^k, \mu\rangle)h^k(x),$$

$$\text{monomials } f^k, h^k : \mathbb{R}^D \to \mathbb{R}, \text{ kernels } g^k : \mathcal{M} \to \mathcal{M} \Big\}.$$

We can see the $\mathcal{G}_D(\mathcal{M})$ can be viewed as a subalgebra of $\mathcal{H}(\mathcal{P}(\mathcal{M})\times\mathcal{M})$. Additionally, we can also see that

- $\mathcal{H}(\mathcal{P}(\mathcal{M}) \times \mathcal{M})$ separates points on $\mathcal{P}(\mathcal{M}) \times \mathcal{M}$. To check this, take $(\mu, x) \neq (\mu', x') \in \mathcal{P}(\mathcal{M}) \times \mathcal{M}$, with either $\mu \neq \mu'$ or $x \neq x'$. If $\mu' \neq \mu$, from Theorem 30.1 by Billingsley (2013), there exists a kernel function $k(x_0, \cdot)$ such that $\int_{\mathcal{M}} k(y, x)(\mu - \mu')(\mathrm{d}x) \neq 0$, otherwise, $\mu = \mu'$. In this case, define $p(\mu, x) = \langle k(x_0, x) \rangle \in \mathcal{H}(\mathcal{P}(\mathcal{M}) \times \mathcal{M})$. If $\mu' = \mu$, $x' \neq x$, let $p(\mu, x) = x$, then $p(\mu, x) \neq p(\mu', x')$. In either case, $\mathcal{H}(\mathcal{P}(\mathcal{M}) \times \mathcal{M})$ separates points on $\mathcal{P}(\mathcal{M}) \times \mathcal{M}$.

- $\mathcal{H}(\mathcal{P}(\mathcal{M}) \times \mathcal{M})$ vanishes at no point on $\mathcal{P}(\mathcal{M}) \times \mathcal{M}$. It can be checked to choose a nonzero constant function as $f_k$ and $h_k$.

Therefore, it follows from the Stone-Weierstrass lemma that $\mathcal{H}(\mathcal{P}(\mathcal{M}) \times \mathcal{M})$ is dense in $\mathcal{C}(\mathcal{P}(\mathcal{M}) \times \mathcal{M})$ with the topology of uniform convergence. Hence, there exists a sequence of functions $p_n, \tilde{p}_n \in \mathcal{H}(\mathcal{P}(\mathcal{M}) \times \mathcal{M})$ such that for any $\epsilon > 0$, there exists $N \in \mathbb{N}$ that for $n \geq N$,

$$\sup_{(\mu,x)\in\mathcal{P}(\mathcal{M})\times\mathcal{M}} \left| p_n(\mu, x) - \partial_{xx}\frac{\delta\Phi}{\delta\mu}(\mu, x) \right| \leq \epsilon, \tag{10}$$

and

$$\sup_{\mu\in\mathcal{P}(\mathcal{M})} \left| \tilde{p}_n(\mu) - \frac{\delta\Phi}{\delta\mu}(\mu, 0) \right| \leq \epsilon. \tag{11}$$

**Step 2**: Let

$$P_n(\mu, x) := \tilde{p}_n(\mu) + \int_0^x \int_0^y p_n(\mu, z)\mathrm{d}z\mathrm{d}y,$$

and

$$\Phi_n(\mu) := \Phi(\delta_0) + \int_0^1 \int_{\mathcal{M}} P_n(\lambda\mu + (1-\lambda)\delta_0, x)(\mu - \delta_0)(\mathrm{d}x)\mathrm{d}\lambda.$$

It can be checked that $\Phi_n \in \mathcal{G}_D(\mathcal{M})$ with polynomial kernels. Now we have

$$P_n(\mu, x) - \frac{\delta\Phi}{\delta\mu}(\mu, x)$$

$$= \tilde{p}_n(\mu) + \int_0^x \int_0^y p_n(\mu, z)\mathrm{d}z\mathrm{d}y -$$

$$\left( \frac{\delta\Phi}{\delta\mu}(\mu, 0) + \int_0^x \int_o^y \partial_{xx}\frac{\delta\Phi}{\delta\mu}(\mu, z)\mathrm{d}z\mathrm{d}y \right)$$

$$= \tilde{p}_n(\mu) - \frac{\delta\Phi}{\delta\mu}(\mu, 0) + \int_0^x \int_0^y \left( p_n(\mu, z)\mathrm{d}z - \partial_{xx}\frac{\delta\Phi}{\delta\mu}(\mu, z) \right) \mathrm{d}z\mathrm{d}y.$$

Thus, by Eq. (10),

$$\sup_{\mathcal{P}(\mathcal{M})\times\mathcal{M}} |\partial_x P_n(\mu, x) - \partial_\mu\Phi(\mu, x)| \leq K\epsilon,$$

$$\sup_{\mathcal{P}(\mathcal{M})\times\mathcal{M}} \left| P_n(\mu, x) - \frac{\delta\Phi}{\delta\mu}(\mu, x) \right| \leq (1 + K^2)\epsilon.$$

Moreover,

$$\Phi_n(\mu) - \Phi(\mu)$$

$$= \left( \Phi(\delta_0) + \int_0^1 \int_{\mathcal{M}} P_n(\lambda\mu + (1-\lambda)\delta_0, x)(\mu - \delta_0)(\mathrm{d}x)\mathrm{d}\lambda \right)$$

$$- \left( \Phi(\delta_0) + \int_0^1 \int_{\mathcal{M}} \frac{\delta\Phi}{\delta\mu}(\lambda\mu + (1-\lambda)\delta_0, x)(\mu - \delta_0)(\mathrm{d}x)\mathrm{d}\lambda \right)$$

$$= \int_0^1 \int_{\mathcal{M}} \left( P_n(\lambda\mu + (1-\lambda)\delta_0, x) - \frac{\delta\Phi}{\delta\mu}(\lambda\mu + (1-\lambda)\delta_0, x) \right)(\mu - \delta_0)(\mathrm{d}x)\mathrm{d}\lambda.$$

Hence,

$$\sup_{\mathcal{P}(\mathcal{M})} |\Phi_n(\mu) - \Phi(\mu)| \leq 2(1 + K^2)\epsilon.$$

Therefore,

$$\|\Phi_n - \Phi\|_{\mathcal{M}} \leq (1 + K + 2(1 + K^2))\epsilon,$$

with $\Phi_n \in \mathcal{G}_D(\mathcal{M})$, which is shown to be dense in $C^{1,1}(\mathcal{M})$.

□

### B.2 Statement and Proof of Wasserstein Lipschitz Continuous

**Lemma B.3** (Wasserstein Lipschitz Continuous). *If Assumption 3.1 holds, then cylindrical function $h(\mu) \in \mathcal{G}_D(\mathcal{M})$ is $C$-Lipschitz continuous according to $\mu \in \mathcal{P}(\mathcal{M})$, i.e., for any measure $\mu, \nu \in \mathcal{P}_2(\mathcal{M})$, there holds*

$$|h(\nu_0) - h(\nu_1)| \leq C\mathcal{W}_2(\nu_0, \nu_1), \tag{12}$$

*where $C$ is a constant.*

*Proof.* Since the kernels $g^d$ are unformly bounded, the input space for outer function $h$ are actually is compact. Therefore, outer function $h : \mathbb{R}^D \to \mathbb{R}$ (a polynomial function) is $L_h$-Lipschitz continuous:

$$|h(z_1) - h(z_2)| \leq L_h\|z_1 - z_2\|_2, \quad \forall z_1, z_2 \in \mathcal{G}, \tag{13}$$

where $\mathcal{G} \subset \mathbb{R}^D$ is a compact subspace. Let $\pi$ be the optimal coupling between $\nu_0$ and $\nu_1$. Then:

$$|h(\nu_0) - h(\nu_1)| \leq L_h \left( \sum_{d=1}^D \left| \langle g^d, \mu_{\nu_0} - \mu_{\nu_1} \rangle_{\mathcal{H}_k} \right|^2 \right)^{1/2}$$

$$\leq L_h\sqrt{D} \max_{1 \leq d \leq D} |\langle g^d, \mu_{\nu_0} - \mu_{\nu_1} \rangle_{\mathcal{H}_k}|.$$

Therefore, we have that

$$|h(\nu_0) - h(\nu_1)|^2 \leq L_h^2 D \max_{1 \leq d \leq D} |\langle g^d, \mu_{\nu_0} - \mu_{\nu_1} \rangle_{\mathcal{H}_k}|^2$$

$$\leq L_h^2 D \max_d \left| \int_{\mathcal{X}} \left( g^d(x) \right)^2 (\mathrm{d}\nu_0 - \mathrm{d}\nu_1)(x) \right|$$

$$\leq L_h^2 D \inf_\pi \max_d \int_{\mathcal{X} \times \mathcal{X}} \left( g^d(x) \right)^2 \mathrm{d}\pi(x, y)$$

$$\leq L_h^2 D L_g^2 \inf_\pi \int_{\mathcal{X} \times \mathcal{X}} \|x - y\|_2^2 \mathrm{d}\pi(x, y)$$

$$= C\mathcal{W}_2(\mu, \nu)^2,$$

where the last inequality follows from the Kantorovich-Rubinstein duality. Therefore, we have that

$$|h(\nu_0) - h(\nu_1)| \leq L_d\sqrt{D}L_g\mathcal{W}_2(\mu, \nu). \tag{14}$$

□

### B.3 Proof of Proposition 3.2

*Proof.* We provide derivation of Proposition 3.2. From Eq. (6), we have the form of $Q^i$. Then, the functional gradient in the form of Fréchet derivative is

$$\nabla_{g^{i,d}} Q^i = \frac{\delta h_{s^i, a^i}}{\delta g} + \sum_{d'=1}^D \frac{\delta(\partial_{d'} h_{s^i, a^i} \langle \nabla g^{i,d'}(x) \cdot \Delta x, \nu^{-i} \rangle)}{\delta g}$$

$$= \partial_d h_{s^i, a^i} \mu_{\nu^{-i}} + \sum_{d'=1}^D \frac{\delta(\partial_{d'} h_{s^i, a^i})}{\delta g} \langle \nabla g^{i,d'}(x) \cdot \Delta x, \nu^{-i} \rangle$$

$$+ \partial_d h_{s^i, a^i} \frac{\delta \langle \nabla g^{i,d}(x) \cdot (\bar{x}^i - x), \nu^{-i}(x) \rangle}{\delta g}.$$

To calculate the last term in $\nabla_{g^{i,d}} Q^i$, we apply the fundamental lemma of calculus of variations. Define function $f(x, g, \nabla g) = g^{i,d}(x) \cdot (\bar{x}^i - x)\nu^{-i}(x)$, then, $\langle \nabla g^{i,d}(x) \cdot (\bar{x}^i - x), \nu^{-i}(x) \rangle$ can be written as

$$\langle \nabla g^{i,d}(x) \cdot (\bar{x}^i - x), \nu^{-i}(x) \rangle$$
$$= \int_{\mathcal{M}} \nabla g^{i,d}(x) \cdot (\bar{x}^i - x)\nu^{-i}(x)\mathrm{d}x$$
$$= \int_{\mathcal{M}} f(x, g, \nabla g)\mathrm{d}x.$$

Therefore, we have that

$$\frac{\delta \langle \nabla g^{i,d}(x) \cdot (\bar{x}^i - x), \nu^{-i}(x) \rangle}{\delta g} = \frac{\partial f}{\partial g} - \nabla \cdot \frac{\partial f}{\partial \nabla g} = -\nabla \cdot ((\bar{x}^i - x)\nu^{-i}(x)).$$

Hence, we have the form in Proposition 3.2.

$$\nabla_{g^{i,d}} Q^i = \partial_d h_{s^i, a^i} \mu_{\nu^{-i}} + \sum_{d'=1}^{D} \frac{\partial_{dd'}^2 h_{s^i, a^i}}{N_i} \sum_{j=1}^{N_i} \nabla g^{i,d'}(x^j) \cdot (\bar{x}^i - x^j)\mu_{\nu^{-i}}$$
$$+ \partial_d h_{s^i, a^i} \nabla \cdot (\nu^{-i}(x)(x - \bar{x}^i))$$

$\square$

### B.4 PROOF OF THEOREM 4.1

*Proof.* Under Assumption 3.1, we know that the cylindrical function $h(\mu)$ is Wasserstein continuous by Lemma B.3. Therefore, we have that

$$|h(\nu_n) - h(\nu_M)| \le C\mathcal{W}_2(\nu_N, \nu_M).$$

Since Wassserstein distance meets the triangle inequality (Panaretos & Zemel, 2019), we have that

$$\mathcal{W}_2(\nu_N, \nu_M) \le \mathcal{W}_2(\nu_N, \nu) + \mathcal{W}_2(\nu_M, \nu).$$

Since the convergence rate of empirical distribution $\nu_N$ to $\nu$ under measure of Wasserstein distance is $O(N^{-1/d})$ (Dudley, 1969), namely,

$$\mathbb{E}[\mathcal{W}_2(\nu_N, \nu)] \le CN^{-1/d}.$$

Therefore, we have that

$$\mathbb{E}[|h(\nu_n) - h(\nu_M)|] \le C\mathbb{E}[\mathcal{W}_2(\nu_N, \nu)] + C\mathbb{E}[\mathcal{W}_2(\nu_M, \nu)]$$
$$\le C_1 N^{-1/d} + C_2 M^{-1/d}.$$

$\square$

### B.5 PROOF OF THEOREM 4.2

*Proof.* First, we prove that the convergence rate of cylindrical function is controlled by the convergence rate of empirical kernel mean embedding.

**Lemma B.4** (Convergence Rate Bound of Kernel Cylindrical Functions (Lemma 5.2, (Venturi & Dektor, 2021))). *Denote the projection of measure $\nu$ on RKHS embedding space $\mathcal{H}_M$ as $\mathcal{P}_D \nu = \sum_d c_d k(x^d, \cdot)$, where $[c_1, \ldots, c_D]^\top =: \boldsymbol{c} = (\boldsymbol{K}_{DD})^{-1}\boldsymbol{b}$ and $b_d = \langle k(x^d, \cdot), \nu \rangle$. We have that $h$ defined in Eq. (4) with one type of kernel converges to $f$ for all $\nu \in \mathcal{P}_2(\mathcal{M})$ with the same rate as $\mathcal{P}_D \nu$ convergences to the kernel mean embedding $\mu_\nu$. Formally, with $\tilde{f} : \mu_\nu \mapsto f(\nu)$, it can be expressed as*

$$|h(\nu) - f(\nu)| \le \sup_\nu \left\| \frac{\delta \tilde{f}}{\delta \mu_\nu} \right\| \|\mu_\nu - \mathcal{P}_D \nu\|_{\mathcal{H}}, \tag{15}$$

*where $\delta \tilde{f}/\delta \mu_\nu$ is the Fréchet derivative of function $\tilde{f}$ and $\mu_\nu$ is the kernel mean embedding defined in Eq. (3).*

From Lemma B.4, the convergence rate of the cylindrical function is controlled by the convergence rate of the empirical kernel mean embedding.

**Lemma B.5** (Convergence Rate of Empirical Kernel Mean Embedding (Theorem 3.4, (Muandet et al., 2017))). *Assume the boundedness for kernel $k$ in Assumption 3.1 holds. Then for any $\delta \in (0, 1)$ with probability at least $1 - \delta$,*

$$\|\mu_\nu - \mathcal{P}_D \nu\|_{\mathcal{H}} \leq \sqrt{\frac{1}{D}} + \sqrt{\frac{2\log(1/\delta)}{D}}. \tag{16}$$

Combining the results from Lemme B.5, we have that the convergence rate of $h$ to $f$ is the multiple of Fréchet derivative and $O(D^{-1/2})$, which proves our results. □

### B.6 PROOF OF THEOREM 4.3

*Proof.* First, we introduce the non-linear two-time-scale stochastic approximation.

**Lemma B.6** (Nonlinear Two-Time-Scale Stochastic Approximation (Borkar, 2008)). *Consider two coupled stochastic approximation processes:*

$$x_{n+1} = x_n + a(n)\left[f(x_n, y_n) + M_n^{(1)}\right], \tag{17}$$

$$y_{n+1} = y_n + b(n)\left[g(x_n, y_n) + M_n^{(2)}\right], \tag{18}$$

*where $x_n \in \mathbb{R}^d$ (slow process) and $y_n \in \mathbb{R}^k$ (fast process), with step sizes $a(n), b(n) > 0$.*

*Assume that*

*(i) $f : \mathbb{R}^d \times \mathbb{R}^k \to \mathbb{R}^d$ and $g : \mathbb{R}^d \times \mathbb{R}^k \to \mathbb{R}^k$ are Lipschitz continuous,*

*(ii) For each fixed $x$, the ODE $\dot{y}(t) = g(x, y(t))$ has a globally asymptotically s equilibrium $y^*(x)$. The ODE $\dot{x}(t) = f(x(t), y^*(x(t)))$ has a globally asymptotically s equilibrium $x^*$,*

*(iii) the sequences $\{a(n)\}$ and $\{b(n)\}$ satisfy Robbins-Monro conditions in Assumption 4.1, and*

*(iv) $\{M_n^{(1)}\}, \{M_n^{(2)}\}$ are martingale differences w.r.t. $\mathcal{F}_n = \sigma(x_m, y_m, M_m^{(1)}, M_m^{(2)}, m \leq n)$, with*

$$\mathbb{E}\left[\|M_n^{(i)}\|^2 \mid \mathcal{F}_n\right] \leq C(1 + \|x_n\|^2 + \|y_n\|^2), \quad i = 1, 2.$$

*Then, the iterates $(x_n, y_n)$ converge almost surely to $(x^*, y^*)$, where $y^* = y^*(x^*)$.*

Base on the Lemma B.6, we rewrite updates of Eq. 7 as stochastic approximation processes:

$$h_{t+1} = h_t + \eta_h\left(F_h(h_t, g_t) + M_h^{t+1}\right), \tag{19a}$$

$$g_{t+1} = g_t + \eta_g\left(F_g(h_t, g_t) + M_g^{t+1}\right), \tag{19b}$$

where $F_h = -\mathbb{E}\left[\frac{\partial\ell}{\partial Q_{\text{tot}}} \cdot \frac{\partial Q_{\text{tot}}}{\partial Q^i}\nabla_h Q^i\right]$ and $F_g$ is defined analogously. $M_h, M_g$ are martingale difference noise terms.

By the SA theory (Borkar, 2008), the updates approximate:

$$\text{(Fast)} \quad \dot{g} = F_g(h, g), \tag{20a}$$

$$\text{(Slow)} \quad \dot{h} = F_h(h, g^*(h)), \tag{20b}$$

where $g^*(h)$ is the equilibrium of Eq. (20a) for fixed $h$.

Since the Bellman operator is a contraction mapping (Littman, 1994), we have that there exists a globally asymptotically s equilibrium $g^*$ and $h^*$ to minimize $\ell$. Therefore, by the Lemma B.6, we have that:

- The fast process Eq. (19b) tracks Eq. (20a), converging to $g^*(h_t)$ for any slow $h_t$.

- The slow process Eq. (19a) converges to $h^*$, which induces $g^* = g^*(h^*)$.

Thus, $(h_t, g_t) \to (h^*, g^*)$ almost surely. □

### B.7 PROOF OF THEOREM 4.4

*Proof.* Theorem 4.4 is the same with Theorem 1 in (Rudi et al., 2015). Define the integral operator $L_k$ for kernel function $k$ by

$$L_k f(x) = \int_{\mathcal{X}} f(s)k(x,s)\mathrm{d}s.$$

For $\lambda > 0$, define the random variable $\mathcal{N}_x(\lambda) = \langle K_x, (L_k + \lambda I)^{-1} K_x \rangle$ with $x \in \mathcal{X}$. The efficient dimension is

$$\mathcal{N}(\lambda) = \mathbb{E}\mathcal{N}_x(\lambda), \quad \mathcal{N}_\infty(\lambda) = \sup_{x \in \mathcal{X}} \mathcal{N}_x(\lambda).$$

**Theorem B.7** (Error Analysis of Nyström Approximation, Theorem 1 (Rudi et al., 2015)). *Under Assumption3.1, 4.2 and 4.3, let $\delta \in (0,1)$, $v = \min(s, 1/2)$, $p = 1 + 1/(2v+\gamma)$ and assume*

$$N_i + M \geq 1655 + 223 \log \frac{6}{\delta} + \left( \frac{38p}{\|L_k\|} \log \frac{114p}{\|L_k\|\delta} \right)^p \tag{21}$$

*Then, the following inequality holds with probability at least $1 - \delta$ for ,*

$$\mathcal{E}(\tilde{g}_{t+1}^{i,d}) \leq \min_{f \in \mathcal{H}} \mathcal{E}(f) + q^2(N_i + M)^{-\frac{2v+1}{2v+\gamma+1}}, \tag{22}$$

*with*

$$q = 6R\left(2\|L_k\| + \frac{C_1}{\sqrt{\|L_k\|}} + \sqrt{\frac{C_2}{\|L_k\|^\gamma}}\right) \log \frac{6}{\delta},$$

$C_1, C_2$ *are constants, and* $\lambda = \|L_k\|(N_i + M)^{-\frac{1}{2v+\gamma+1}}$ *and* $L \geq \max(67, 5\mathcal{N}_\infty(\lambda)) \log \frac{12}{\lambda\delta}$.

$\square$

In our scenario, for a large swarm with batch size, the $N_i + M$ will easy meet the assumption in Theorem B.7. For example, if a swarm of $N = 32$ with batch size $B = 128$, along with kernel number $M = 64$, $N_i + M = B \cdot N + M$ will be 4160, which may statisfy the assumption with certain $\delta$.

## C APPENDED REMARKS

### C.1 REMARKS ON KERNEL CYLINDRICAL FUNCTIONS AND MEAN FIELD EMBEDDING

**Remarks C.1** (Requirements on kernel by Lipschitz continuity). *The Lipschitz continuity requirement limits the choice of kernel functions. Such as*

- ***Polynomial kernels**: $k(y,x) = (\alpha x \cdot y + c)^d$ violates the condition when input space $\mathcal{X}$ is unbounded, as the gradients grow polynomially with $\|x\|_2$.*

- ***Sigmoid kernels**: $k(y,x) = \tanh(\alpha x \cdot y + c)$ could fail to satisfy global Lipschitz continuity due to saturation effects in nonlinear regions.*

- ***Gaussian kernels**: $k(y,x) = \exp(-\gamma \|x - y\|_2^2)$ generally meet the requirement with $L_g = \gamma \sup_x \|x\|_2/2$.*

**Remarks C.2** (Inner Product between mean-field measure and component functions). *The inner product between mean field measure and component function $g^{i,d}$ evaluates to:*

$$\langle g^{i,d}, \mu_{\nu^{-i}}^d \rangle = \frac{1}{N_i} \sum_{m=1}^{M} \sum_{j=1}^{N_i} \alpha_m^d k^d(x^m, x^j) = \frac{\mathbf{1}^\top \boldsymbol{K}^d \boldsymbol{\alpha}^d}{N_i}, \tag{23}$$

*where $\boldsymbol{K}^d \in \mathbb{R}^{N_i \times M}$ is the Gram matrix with $\boldsymbol{K}_{jm}^d = k^d(x^j, x^m)$ and $\mathbf{1} \in \mathbb{R}^{N_i}$ is an all-ones vector.*

Table 6: Kernel Functions and Corresponding Partial Derivative

| Kernel Type | Kernel $k(y, x)$ | Gradient of kernel $\partial_x k(y, x)$ |
|---|---|---|
| Linear | $x \cdot y + c$ | $y$ |
| Polynomial | $(\alpha x \cdot y + c)^d$ | $\alpha d(\alpha x \cdot y + c)^{d-1} y$ |
| Gaussian | $\exp(-\gamma\|x - y\|^2)$ | $-2\gamma(x - y)\exp(-\gamma\|x - y\|^2)$ |
| Laplacian | $\exp(-\gamma\|x - y\|_1)$ | $-\gamma\mathrm{sign}(x - y)\exp(-\gamma\|x - y\|_1)$ |
| Sigmoid | $\tanh(\alpha x \cdot y + c)$ | $\alpha y(1 - \tanh^2(\alpha x \cdot y + c))$ |

## D  REMARKS ON KERNEL FUNCTIONS

We list several kernels frequently appearing in the literature.

In our work, in consideration of Lipischitz continuity, representation capability and easy to calculate, we adopt polynomial and Gaussian kernels.

### D.1  REMARKS ON MEAN-FIELD REPRESENTATION OF VALUE FUNCTIONS

**Remarks D.1** (Expansion of Eq. (6)). *Eq. 6 is expanded as:*

$$Q^i(\boldsymbol{s}, \boldsymbol{a}) = h_{s^i, a^i}\left(\frac{\mathbf{1}^\top \boldsymbol{K}^1 \boldsymbol{\alpha}^1}{N_i}, \ldots, \frac{\mathbf{1}^\top \boldsymbol{K}^D \boldsymbol{\alpha}^D}{N_i}\right)$$

$$+ \frac{1}{N_i} \sum_{d=1}^D \partial_d h_{s^i, a^i} \sum_{m=1}^M \sum_{j=1}^{N_i} \alpha_m^d \partial_x k^d(x^m, x^j) \cdot (\bar{x}^i - x^j).$$

**Remarks D.2** (Mean field representation of state value function and advantage funcion). *Similarly, we can present the state value function $V^i(\boldsymbol{s})$ and advantage function $A^i(\boldsymbol{s}, \boldsymbol{a})$ with the mean field representation in Eq. (6) as*

$$V^i(\boldsymbol{s}) = h_{s^i}^v\left(\langle g_v^{i,1}, \mu_{\nu^{-i}}\rangle, \ldots, \langle g_v^{i,D}, \mu_{\nu^{-i}}\rangle\right) + \sum_{d=1}^D \partial_d h_{s^i}^v \langle \nabla g_v^{i,d}(x) \cdot \Delta x, \nu^{-i}\rangle,$$

*and*

$$A^i(\boldsymbol{s}) = h_{s^i, a^i}^{adv}\left(\langle g_{adv}^{i,1}, \mu_{\nu^{-i}}\rangle, \ldots, \langle g_{adv}^{i,D}, \mu_{\nu^{-i}}\rangle\right) + \frac{1}{N_i} \sum_{d=1}^D \partial_d h_{s^i, a^i}^{adv} \langle \nabla g_{adv}^{i,d}(x) \cdot \Delta x, \nu^{-i}\rangle,$$

*where $h_{s^i}^v$ and $h_{s^i, a^i}^{adv}$ are the cylindrical kernel functions, with kernel functions $\{g_v^{i,d}\}$ and $\{g_{adv}^{i,d}\}$ for value function $V$ and advantage function $A$, respectively. In this paper, we focus on the $Q$ function, while we think it is also interesting to expand our conclusions to value and advantage functions.*

**Remarks D.3** (Explicit form of Fréchet derivative). *In discrete particle approximation with $N_i$ neighbors, Eq. (8) is:*

$$\nabla_{g^{i,d}} Q^i = \sum_{j=1}^{N_i}\left[\frac{\partial_d h}{N_i} + \sum_{d'} \frac{\partial_{dd'}^2 h}{N_i^2} \sum_{j'} \nabla g^{d'}(x^{j'}) \Delta x^{j'}\right] k^d(x^j, \cdot) + \frac{\partial_d h}{N_i} \sum_{j=1}^{N_i} \left[\delta_{x^j} - \nabla \delta_{x^j} \cdot \Delta x^j\right].$$

### D.2  REMARKS ON NYSTRÖM APPROXIMATION

**Remarks D.4.** *The gradient inner product admits explicit computation:*

$$\langle k(x^n, \cdot), \nabla_{g^{i,d}} Q^i\rangle = \sum_{j=1}^{N_i}\left[\frac{2\partial_d h}{N_i} + \sum_{d'} \frac{\partial_{dd'}^2 h}{N_i^2} \sum_{j'} \nabla g^{d'}(x^{j'}) \cdot \Delta x^{j'}\right] k^d(x^n, x^j)$$

$$- \frac{\partial_d h}{N_i} \sum_{j=1}^{N_i} \nabla_x k^d(x^n, x^j) \cdot \Delta x^j \tag{24}$$

**Remarks D.5** (Anchor Point Selection). *There are several principled ways to choose anchor points $\{z^l\}_{l=1}^{L}$:*

- ***Random Subsampling***: *Select $L$ points uniformly from RKHS anchor points $\{x^n\}_{n=1}^{N_i+M}$ in $g_{t+1}^{i,d}$.*

$$z^l \sim Uniform(\{x^n\}_{n=1}^{N_i+M}), \quad l = 1, ..., L.$$

  *Pros: $O(1)$ computational cost. Cons: May miss important regions.*

- ***k-means Centers***: *Solve*

$$\{z^l\} = \arg\min_{\{c_l\}} \sum_{x \in \{x_m\}} \min_{1 \le l \le L} \|x - c_l\|^2.$$

  *Pros: Captures data geometry. Cons: $O(N_i L T)$ computation complexity for $T$ iterations.*

- ***Kernel Herding***: *Select points maximizing the minimum kernel similarity:*

$$z_{l+1} = \arg\max_{x \in \{x_m\}} \sum_{l'=1}^{l} k(x, z_{l'}) - \frac{2}{N_i} \sum_{j=1}^{N_i} k(x, x^j).$$

  *Pros: Constructs maximally representative points. Cons: $O(N_i L T)$ computation complexity for $T$ iterations.*

- ***Leverage Score Sampling***: *Sample with probability proportional to diagonal entries of the kernel matrix:*

$$p_j = \frac{(K_{MM})_{jj}}{tr(K_{MM})}, \quad z^l \sim p_j.$$

  *Pros: Preserves spectral structure of the RKHS.*

*In this paper, we apply the random subsampling method for simplicity.*

# E  IMPLEMENTATION DETAILS OF SKARL AND BASELINES

## E.1  IMPLEMENTATION DETAILS OF SKARL

**Base Algorithm of Credit Assignment for SKARL**  We apply VDN (Sunehag et al., 2017) as the basic credit assignment algorithm for SKARL. Namely, the total $Q_{\text{tot}}$ value is calculated by

$$Q_{\text{tot}}(\boldsymbol{s}, \boldsymbol{a}) = \sum_{i=1}^{N} Q^i(\boldsymbol{s}, \boldsymbol{a}).$$

**Kernel Cylindrical Function Implementation**  We adopt a hypernetwork (Ha et al., 2016) for kernel cylindrical function network. Namely, the ego state and action $(s^i, a^i)$ are used to generate the parameters of a network for processing $\mu_{\nu_{N_i}}$.

**Tricks**  We apply several tricks to help stabilize and fasten training.

- **Dual Network Update**: To avoid over-estimation of $Q$ value, we apply double Q learning framework (Van Hasselt et al., 2016).
- **Entropy Regularization**: To avoid the performance drops in the last epochs during training, we apply entropy regularization on the actor policy.

**Codebase**  We apply SKARL and baselines with Jax. We organize the code in JaxMARL (Rutherford et al., 2023) for better organization and class inheritance. We plan to release full codes afterwards. For now, the code for important implementation can be found via anonymous Github link: https://anonymous.4open.science/r/SKARL-050D.

**Hyperparameters**  In this paragraph, we list the hyperparameters in 7 and 8.

Table 7: Environment & Training Configuration

| Environment | | Training | | Optimizer | |
|---|---|---|---|---|---|
| Hyperparameter | Value | Hyperparameter | Value | Hyperparameter | Value |
| Agent Number | 4 / 16 / 64 | Total Time Steps | 2M | Learning Rate | 7e-4 |
| Environments Number | 128 | Update Steps Number | 50 | Max Grad Norm | 10 |
| Test Environment Number | 8 | Target Update Interval | 8 | Optimizer | ADAM |
| Max Train Env Timesteps | 50 | Test Interval | 50k | EPS | 1e-5 |
| Max Test Env Timesteps | 100 | | | Weight Decay | 0 |
| Buffer | | Exploration | | Learning rate Decay | |
| Hyperparameter | Value | Hyperparameter | Value | Hyperparameter | Value |
| Buffer Size | 8192 | Epsilon | $1.0 \rightarrow 0.05$ | $\eta_h$ | $1/t^{0.6}$ |
| Buffer Batch Size | 32 | Epsilon Anneal Time | 50k | $\eta_g$ | $1/t^{0.8}$ |
| Buffer Sample | Uniform | Anneal Method | Linear | Basic LR | 7e-5 |

Table 8: Network & Algorithm Architecture

| Network | | Algorithm | |
|---|---|---|---|
| Hyperparameter | Value | Hyperparameter | Value |
| Embedding Net Layer | 3 | TD Lambda | 0.95 |
| Agent Hidden Dim | 16 | Gamma | 0.99 |
| Mixer Embedding Dim | 256 | Entropy Rate | 0.5 |
| Mixer Hypernet Hidden Dim | 256 | Anchor Points Number | $L = 64$ |
| Attention Dim | 64 | Tikhonov Coefficient | 0.5 |
| Activation | ReLU | Polynomial Kernel | $(\alpha,d,c)=(1,2,1),(1,3,1)$ |
| FC Init Scale | 2.0 | Gaussian Kernel | $\gamma = 0.5, 1.0$ |

### E.2 COMMON SETTINGS FOR ENVIRONMENT

For learning stability and environment consistency, we conduct following tricks:

**Re-scale of Environment**   To make environment scalable, we conduct re-scale of world size of environment according to the agents as below:

$$\text{world size} = 2 * \min(\sqrt{N} - 1, 1),$$

where world size serves as the boundary value of environment as $[-\text{world size}, \text{world size}] \times [-\text{world size}, \text{world size}]$ and $N$ denotes the number of agents.

**Reset of Agents and Landmarks**   We generate the new agents and landmark uniformly in the world of environment, namely, $p^i \sim \text{Uniform}([-\text{world size}, \text{world size}] \times [-\text{world size}, \text{world size}])$ for $i \in \{1, \dots, N\}$. In some implementations, a reject sampling is adopted to avoid collision between generated agents and landmarks (such as codebase of InforMARL (Nayak et al., 2023), JaxMARL (Rutherford et al., 2023), Mava (de Kock et al., 2023) and so on). However, we do not adopt such rejection, due to the consideration of time consumption. Instead, we separate the environment world into grids and sample among grids to avoid collision.

During both training and evaluation phases in the **Target** and **Coverage** environments, the episode terminates and resets automatically once all agents successfully reach their assigned goals (or all landmarks are uniquely covered for the Coverage task). This design ensures episodic training and prevents infinite loops. However, since agents are able to receive one-time rewards for several times, the total episodic reward may temporarily exceed the theoretical maximum (e.g., $N \times 10$ for $N$ agents) during resets due to reward accumulation in the final timestep.

**Size and Velocity Settings of Agents and Landmarks**   The settings for agents and landmarks are listed as below in 9.

Table 9: Environment Setup

| Hyperparameter | Value |
|---|---|
| Agent Size | 0.15 |
| Landmark Size | 0.225 |
| Agent Maximum Speed | 0.65 (Move)
N/A (Target/Coverage) |
| Agent Acceleration | 5 (Move)
2 (Target/Coverage) |

# F  ADDITIONAL EXPERIMENTS

## F.1  EXPERIMENTS IN TARGET ENVIRONMENT

In this section, we provide the analysis of results for task **Target**. The experimental results in the Target environment demonstrate SKARL's ability to maintain task performance while balancing safety and scalability across different swarm sizes.

For small swarms ($N = 4$), SKARL achieves near-optimal performance with a global reward of 329.3, comparable to QMIX (337.0) and QPLEX (330.3), while ensuring a 100% success rate. However, it exhibits a higher collision count ($7.2 \pm 3.15$) compared to QMIX ($0.67 \pm 0.35$) and QPLEX ($1.3 \pm 0.982$), suggesting a trade-off between task completion and collision avoidance in simpler settings.

As the swarm scales to $N = 16$, SKARL significantly outperforms value-based methods (QMIX, QPLEX, MFRL), which suffer from catastrophic reward degradation (e.g., QPLEX: $-3.1 \times 10^4$). Although MAPPO achieves a higher reward (12.0), its success rate drops to 40.6%, whereas SKARL maintains a 100% success rate despite increased collisions ($23.2 \pm 20.5$). Additionally, SKARL reduces collisions by 32% compared to MFRL, indicating its robustness in mid-scale coordination, which aligns with findings from the Move environment in 2.

In large-scale swarms ($N = 64$), SKARL demonstrates superior scalability, achieving a positive reward (44.75) while all baselines fail (rewards $\leq 0$). Notably, while the collision count remains high ($44.3 \pm 10.6$), the drastic improvement in reward over MFRL ($-5.7 \times 10^5$) and QMIX ($-6.4 \times 10^5$) suggests that SKARL effectively prevents catastrophic failures in complex scenarios. The low success rate (3.1%) implies that further optimization is needed for very large swarms, but the results highlight SKARL's ability to maintain functional performance where other methods collapse.

Overall, SKARL exhibits strong scalability in the Target environment, particularly excelling in maintaining task success and reward stability as swarm size increases, with a trade-off in collision avoidance at larger scales. This aligns with its performance in the Move environment, where it achieves a 96% collision reduction at $N = 64$, reinforcing its effectiveness in large-scale multi-agent coordination. However, the problem of scaling up in Target environment remains to be solved, which require further works.

## F.2  EXPERIMENTS IN COVERAGE ENVIRONMENT

Table 10: Performance Comparison between SKARL and Baselines in Coverage Environment

| Algorithm | $N = 4$ | | | | $N = 16$ | | | | $N = 64$ | | | |
|---|---|---|---|---|---|---|---|---|---|---|---|---|
| | R($\uparrow$) | T($\downarrow$) | # col($\downarrow$) | S($\uparrow$) | R($\uparrow$) | T($\downarrow$) | # col($\downarrow$) | S($\uparrow$) | R($\uparrow$) | T($\downarrow$) | # col($\downarrow$) | S($\uparrow$) |
| MAPPO | 339.6 | 0.40 | $0.26_{\pm 0.561}$ | $\mathbf{1.00}_{\pm 0.0}$ | 167.6 | 0.57 | $5.3_{\pm 2.72}$ | $0.13_{\pm 0.562}$ | 97.3 | 0.87 | $18.4_{\pm 8.35}$ | $0.05_{\pm 0.009}$ |
| MFRL | $\mathbf{396.6}$ | 0.52 | $\mathbf{0.03}_{\pm 0.0}$ | $\mathbf{1.00}_{\pm 0.0}$ | 187.0 | 0.62 | $2.5_{\pm 1.65}$ | $0.12_{\pm 0.456}$ | 216.2 | 0.86 | $15.1_{\pm 2.32}$ | $0.04_{\pm 0.871}$ |
| QMIX | 275.4 | 0.39 | $4.94_{\pm 2.46}$ | $\mathbf{1.00}_{\pm 0.0}$ | 259.5 | $\mathbf{0.52}$ | $19.5_{\pm 5.3}$ | $0.19_{\pm 0.76}$ | 324.2 | 0.92 | $\mathbf{11.8}_{\pm 4.13}$ | $0.10_{\pm 0.526}$ |
| QPLEX | 318.5 | $\mathbf{0.38}$ | $0.56_{\pm 0.194}$ | $\mathbf{1.00}_{\pm 0.0}$ | 298.7 | 0.61 | $7.3_{\pm 6.22}$ | $0.21_{\pm 0.512}$ | 834.5 | 0.85 | $21.5_{\pm 3.65}$ | $0.14_{\pm 0.290}$ |
| SKARL | 387.2 | 0.51 | $0.15_{\pm 0.870}$ | $\mathbf{1.00}_{\pm 0.0}$ | $\mathbf{320.8}$ | 0.61 | $\mathbf{2.42}_{\pm 1.67}$ | $\mathbf{0.22}_{\pm 0.342}$ | $\mathbf{907.3}$ | $\mathbf{0.76}$ | $15.3_{\pm 5.37}$ | $\mathbf{0.17}_{\pm 0832}$ |

Table 11: Flexibility Performance of SKARL in Coverage Environment

| Training | Metric | $M=4$ | $M=16$ | $M=64$ | $M=128$ | $M=256$ |
|---|---|---|---|---|---|---|
| $N=4$ | R/$N$ | 96.8 | 23.7 | 0.3 | -1.2 | -9.2 |
| | T (step) | 51 | 74 | 92 | 100 | 100 |
| | (# col)/$N$ | 0.0375 | 0.76 | 6.932 | 32.4 | 78.9 |
| | S% | 100 | 72 | 4 | 0 | 0 |
| $N=16$ | R/$N$ | 97.5 | 24.05 | 22.3 | 4.3 | 0.82 |
| | T (step) | 43 | 61 | 67 | 94 | 100 |
| | (# col)/$N$ | 0.0457 | 0.19 | 0.203 | 2.54 | 5.21 |
| | S% | 100 | 79 | 6 | 6.25 | 0 |
| $N=64$ | R/$N$ | 96.2 | 25.8 | 14.2 | 9.3 | 3.52 |
| | T (step) | 41 | 56 | 76 | 89 | 92 |
| | (# col)/$N$ | 0.0557 | 0.285 | 0.239 | 0 | 9.68 |
| | S% | 100 | 84 | 13 | 75 | 5 |

Table 12: Performance Comparison between SKARL and Baselines in Line Environment

| Algorithm | $N=4$ | | | | $N=16$ | | | | $N=64$ | | | |
|---|---|---|---|---|---|---|---|---|---|---|---|---|
| | R($\uparrow$) | T($\downarrow$) | # col($\downarrow$) | S($\uparrow$) | R($\uparrow$) | T($\downarrow$) | # col($\downarrow$) | S($\uparrow$) | R($\uparrow$) | T($\downarrow$) | # col($\downarrow$) | S($\uparrow$) |
| MAPPO | 422.3 | 0.31 | $0.10_{\pm 0.20}$ | $1.00_{\pm 0.00}$ | 563.4 | 0.43 | $1.50_{\pm 0.90}$ | $0.30_{\pm 0.20}$ | 1462.7 | 0.72 | $8.00_{\pm 3.00}$ | $0.22_{\pm 0.08}$ |
| MFRL | 444.8 | 0.25 | $0.05_{\pm 0.10}$ | $1.00_{\pm 0.00}$ | 591.2 | 0.43 | $0.90_{\pm 0.60}$ | $0.36_{\pm 0.18}$ | 1604.3 | 0.68 | $6.00_{\pm 2.50}$ | $0.27_{\pm 0.09}$ |
| QMIX | 421.6 | 0.25 | $0.12_{\pm 0.25}$ | $1.00_{\pm 0.00}$ | 572.1 | 0.49 | $1.80_{\pm 1.10}$ | $0.32_{\pm 0.19}$ | 1510.4 | 0.64 | $7.20_{\pm 2.80}$ | $0.24_{\pm 0.09}$ |
| QPLEX | **449.7** | **0.27** | $0.07_{\pm 0.15}$ | $1.00_{\pm 0.00}$ | 608.0 | **0.42** | $1.20_{\pm 0.70}$ | $0.38_{\pm 0.17}$ | 1624.9 | 0.67 | $6.50_{\pm 2.60}$ | $0.26_{\pm 0.09}$ |
| SKARL | 418.9 | 0.23 | $\mathbf{0.03}_{\pm 0.08}$ | $1.00_{\pm 0.00}$ | **615.6** | 0.41 | $\mathbf{0.70}_{\pm 0.50}$ | $\mathbf{0.40}_{\pm 0.16}$ | **1765.8** | **0.66** | $\mathbf{5.50}_{\pm 2.20}$ | $\mathbf{0.30}_{\pm 0.10}$ |

## F.3 EXPERIMENTS IN LINE ENVIRONMENT

## F.4 ABLATION STUDY

**Is it necessary to apply gradient in RKHS?** There is another way to conduct gradient for cylindrical function: directly update in the Euclidean space (Schwenker et al., 2001). Here we provide a comparison with this method with $N=4$ and kernel number is $64$ in Move environment in Figure 3. The result indicates that with RKHS gradient, both the training stability and final performance are improved.

**How number of anchors affect the result?** We compare the performance of different anchor points number $L=1,2,8,32$ under Move task with agent number $N=4$. As is demonstrated in Figure 4, more anchor points only help to stabilize the training process (as the performance of $L=32$ achieves the most stale training curve), while the convergence speed and final performance is scarcely affected. Furthermore, since full performance can be achieved with anchor points number $1$, it is indicated that SKARL can apply at least one kernel number $L$ with $L \leq \sqrt{N}$ to achieve lower computation complexity compared with value decompostion algorithms e.g. QMIX (as discussed in Section 4).

**How types of kernels affect the result?** We compare specific choices of different kernels under Move task with agent number $N=4$. Specificly, we compare the choice of Gaussian kernel and polynomial kernel. For the Gaussian kernel, we adopt $\gamma$ as $(0.5, 1.0, 2.0)$ and for polynomial kernel, we set parameters as $(\alpha, d, c) = (1, 2, 1), (1, 3, 1), (1, 4, 1)$. The results are demonstrated in Figure 5. We conclude that the choice of kernels may not affect the final performance, as long as the representation capability of this kernel is strong enough.

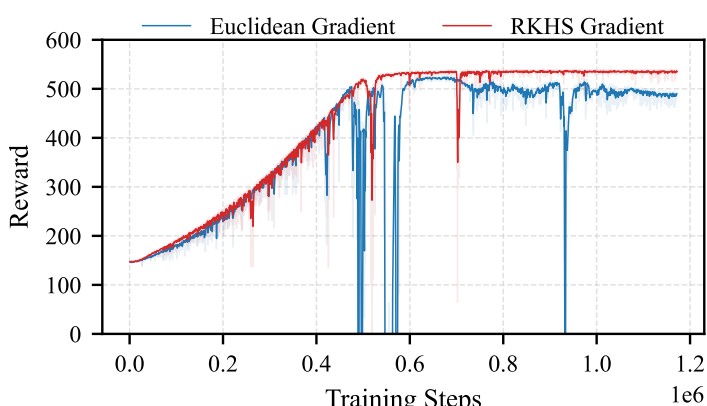

Figure 3: Comparison between gradient in RKHS space and Euclidean space.

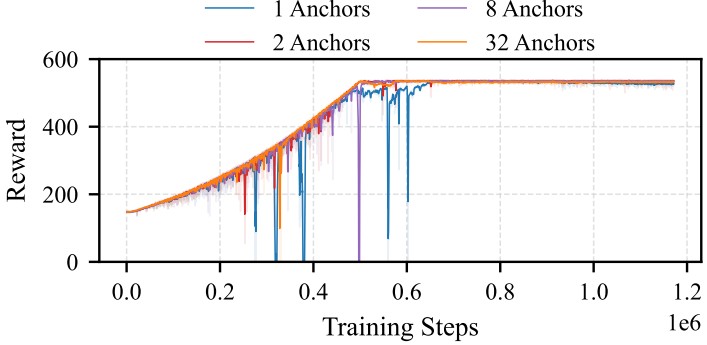

Figure 4: Comparison between different number of anchor points.

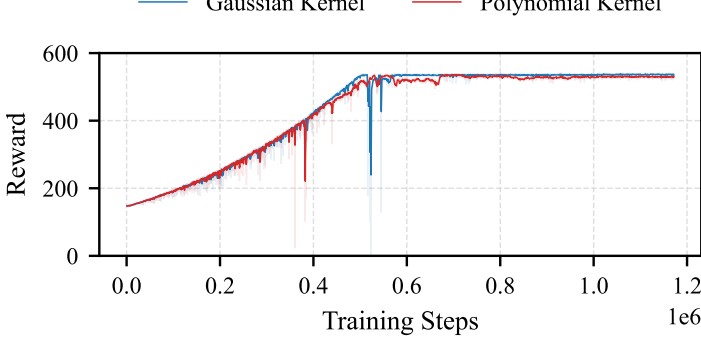

Figure 5: Comparison between different number of kernel types.

Table 13: Flexibility Performance of SKARL in Line Environment

| Training | Metric | $M = 4$ | $M = 16$ | $M = 64$ | $M = 128$ | $M = 256$ |
|---|---|---|---|---|---|---|
| $N = 4$ | R/$N$ | 104.7 | 32.2 | -4.3 | -10.5 | -36.4 |
| | T (step) | 23 | 54 | 87 | 100 | 100 |
| | (# col)/$N$ | 0.0075 | 0.076 | 0.950 | 4.1 | 12.1 |
| | S% | 100 | 72 | 8 | 0 | 0 |
| $N = 16$ | R/$N$ | 117.5 | 38.5 | 20.4 | 6.3 | 0.72 |
| | T (step) | 32 | 41 | 84 | 91 | 100 |
| | (# col)/$N$ | 0.0005 | 0.044 | 0.103 | 0.874 | 1.54 |
| | S% | 100 | 40 | 24 | 3.25 | 0 |
| $N = 64$ | R/$N$ | 123.2 | 53.4 | 27.58 | 18.9 | 2.31 |
| | T (step) | 21 | 31 | 66 | 77 | 82 |
| | (# col)/$N$ | 0.0002 | 0.029 | 0.085 | 0.376 | 0.985 |
| | S% | 100 | 84 | 30 | 27 | 18 |

Table 14: Performance Comparison between different value decomposition methods.

| Algorithm | $N = 4$ | | | $N = 16$ | | | $N = 64$ | | |
|---|---|---|---|---|---|---|---|---|---|
| | R(↑) | # col(↓) | S(↑) | R(↑) | # col(↓) | S(↑) | R(↑) | # col(↓) | S(↑) |
| SKARL | 902.8 | $\mathbf{0}_{\pm 0}$ | $0.15_{\pm 0.0192}$ | 3755.9 | $12.32_{\pm 5.847}$ | $\mathbf{0.17}_{\pm 0.0500}$ | 14423.8 | $7.9_{\pm 5.37}$ | $0.15_{\pm 0.0334}$ |
| SKARL-QMIX | 921.2 | $\mathbf{0}_{\pm 0}$ | $0.15_{\pm 0.0102}$ | 3857.2 | $10.23_{\pm 8.421}$ | $\mathbf{0.18}_{\pm 0.0431}$ | 14512.3 | $6.2_{\pm 4.32}$ | $0.15_{\pm 0.781}$ |
| SKARL-QPLEX | $\mathbf{922.7}$ | $\mathbf{0}_{\pm 0}$ | $0.15_{\pm 0.0021}$ | $\mathbf{3920.1}$ | $\mathbf{9.42}_{\pm 3.412}$ | $\mathbf{0.18}_{\pm 0.0622}$ | $\mathbf{14589.1}$ | $7.9_{\pm 2.98}$ | $\mathbf{0.16}_{\pm 0.676}$ |

**How does anchor points distribute?** We plot the distribution of anchor points with UMAP in Figure 6 with $N = 4$. We can see the anchor points of Gaussian kernel follows nearly a uniform distribution, while anchor points of polynomial kernel follows certain pattern.

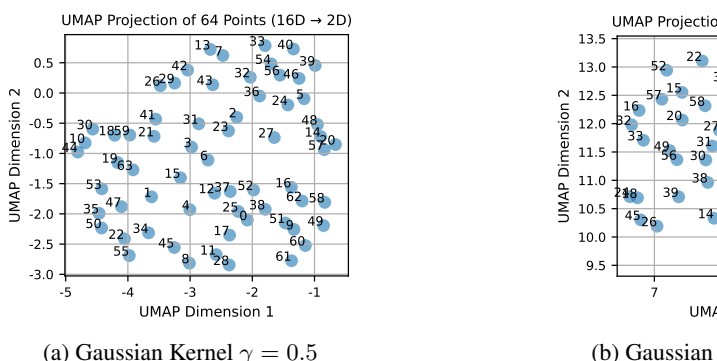

(a) Gaussian Kernel $\gamma = 0.5$      (b) Gaussian Kernel $\gamma = 1.0$

Figure 6: Gaussian kernel anchor points distribution.

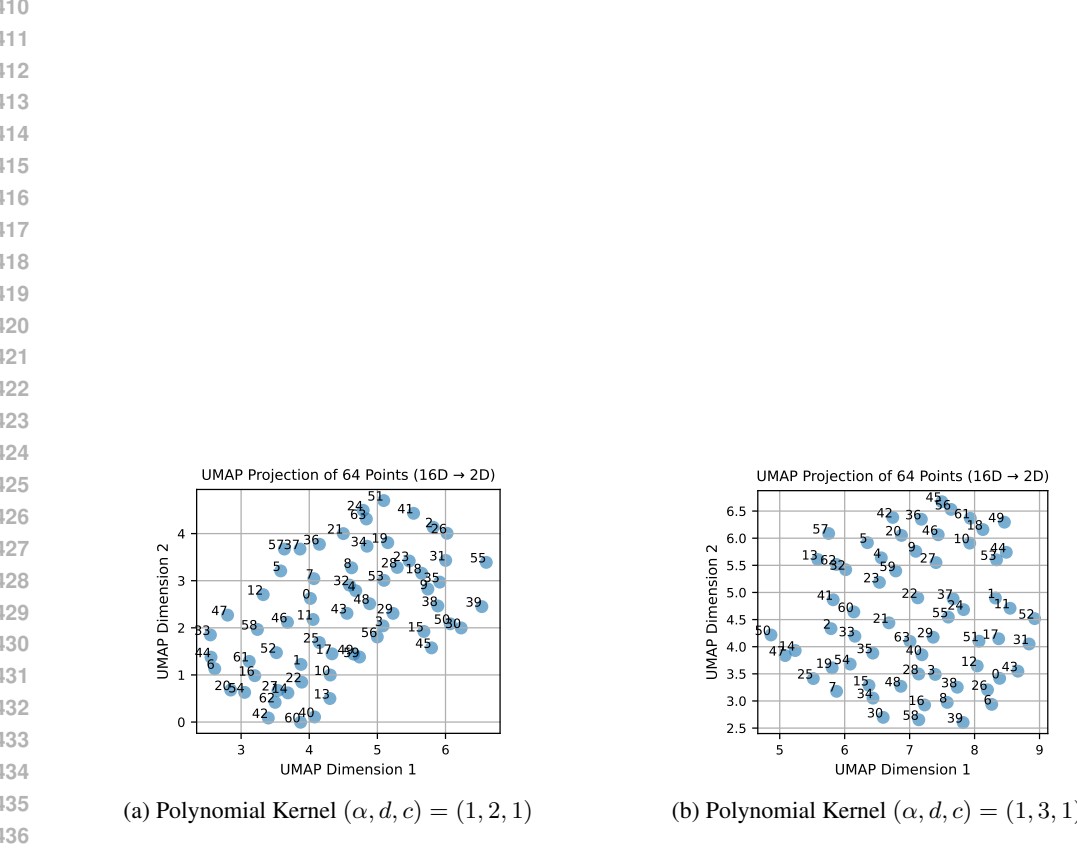

(a) Polynomial Kernel $(\alpha, d, c) = (1, 2, 1)$        (b) Polynomial Kernel $(\alpha, d, c) = (1, 3, 1)$

Figure 7: Polynomial kernel anchor points distribution.

