# OpenReview forum: "SKARL: Provably Scalable Kernel Mean Field Reinforcement Learning for Variable-Size Multi-Agent Systems"
_ICLR.cc/2026/Conference — Submitted to ICLR 2026_

### Official Review · Reviewer_oYza · 2025-10-27

**Soundness:** 3
**Presentation:** 2
**Contribution:** 3
**Rating:** 4
**Confidence:** 3

**Summary:**

This work introduces SKARL (Scalable Kernel MeAn-Field Multi-Agent Reinforcement Learning), a novel framework that integrates mean-field learning with reproducing kernel Hilbert space (RKHS) representations. By using kernel mean embeddings, SKARL provides agents with a rich, high-dimensional feature representation of the entire population distribution, rather than a coarse statistical summary. The authors develop an efficient algorithm using functional gradients and Nyström approximations, proving theoretically that their method is both highly expressive (a universal approximator) and robust to changes in the population (Wasserstein-Lipschitz continuous). Empirically, SKARL outperforms strong MARL baselines in large-scale cooperative tasks.

**Strengths:**

1. By representing the swarm as a distribution instead of a fixed set of agents, a policy trained with one number of agents (e.g., 64) can be directly deployed in environments with a different number of agents (e.g., 4 to 256) without any retraining. This overcomes a major limitation of traditional methods.

2. Unlike previous mean-field methods that use simple statistics (like averages), SKARL uses kernel mean embeddings in a reproducing kernel Hilbert space (RKHS). This provides a much richer, high-dimensional representation of the agent population, allowing the system to capture complex structural information and higher-order differences between distributions.

3. In experiments, SKARL is shown to achieve superior performance on large-scale cooperative tasks, "consistently outperforming strong MARL baselines" in both cumulative reward and training stability.

**Weaknesses:**

1. The authors claim that existing mean-field approaches rely on first-order moment statistics, which provide only coarse summaries of the population. This simplification limits expressiveness and hinders adaptation across swarm sizes, since higher-order structural differences between distributions are ignored. I suggest the authors provide more theoretical or empirical discussion on this limitation instead of directly presenting their solutions, which would further strengthen the contributions of this work. For example, why is higher-order information valuable? Why do previous mean-field approaches limit expressiveness?

2. This work improves the previous mean-field Q-function and embeds interactions using a kernel cylindrical representation. The Q-function is equal to the polynomial combinations of KME kernels. This combination increases the computational complexity compared to previous mean-field methods. Does this method scale to a large number of agents (from a kernel computational complexity perspective)?

3. The authors claim that the representation method integrates seamlessly with standard multi-agent value-decomposition methods such as VDN (Sunehag et al., 2017), QMIX (Rashid et al., 2018), and QPLEX (Wang et al., 2020). However, they do not incorporate the representation method into these methods. It is unclear whether the representation method can improve the performance of current value-decomposition methods.

**Questions:**

Please see the Weaknesses above.

---

> ### Author Response · Authors · 2025-12-03
> **Response**
>
> Thank you for your thoughtful comments. We appreciate the reviewer’s detailed feedback and the opportunity to provide further clarifications. Below are our responses to the weaknesses and questions raised:
>
> # Weakness
> ## Weakness 1
> The first-order moment statistics (such as means) used in traditional mean-field reinforcement learning (MFRL) are coarse approximations because it overlooks important nonlinear interactions between agents, especially as the swarm size grows. On the other hand, higher-order statistics, such as variances, covariances, and higher moments, are valuable because they allow the model to capture more complex interactions between agents. For example, while the mean only captures the central tendency of the population, higher-order moments can reveal how agents deviate from the average, providing insight into the variance or correlations within the population. This is crucial for capturing more nuanced behaviors in larger swarms. Traditional MFRL, relying on first-order moments, ignores these higher-order structural differences, making it less expressive and adaptable to larger agent populations. As swarm sizes increase, the assumptions behind first-order methods break down because they cannot account for the complex dependencies between agents that emerge in larger, more heterogeneous populations.
>
> In the revised manuscript, we added a discussion above the method part, clearly articulating how kernel mean embeddings and cylindrical kernel function improves the expressiveness.
>
> ## Weakness 2
> We would like to point out that the computational complexity of our approach has already been analyzed in Section 4 of the manuscript. In particular, we have provided a detailed comparison of the computational complexity and key metrics between our method (SKARL), QMIX, and MFRL in Table 1. As shown in the table, the complexity of the Q function in our approach (SKARL) scales as $O(B(L^2N+L^3)D)$ where $L$ is the number of landmark points for each $g^d$, $N$ is the number of agents, and $D$ is the number of kernel features. This reflects the additional complexity introduced by the kernel embeddings, which require computation of inner products among agents’ interactions. While this increases the complexity compared to simpler methods like MFRL (which scales linearly), it still compares favorably to QMIX, where the complexity grows exponentially with the number of agent.
>
> ## Weakness 3
> We would like to clarify that our method is based on the calculation of individual Q-functions for each agent, while value-decomposition methods such as VDN, QMIX, and QPLEX focus on aggregating these individual Q-functions into a global Q-function. In our approach, we do integrate with VDN, as mentioned in Appendix E.1. Specifically, our method calculates the individual Q-functions for each agent, and these Q-functions are then aggregated by the VDN framework to compute the global Q-function. This combination allows our kernel-based representation to enrich the individual Q-functions while still benefiting from the VDN approach to value decomposition.
>
> In response to your suggestion, we have added new results in the revised manuscript demonstrating how SKARL can be combined with QMIX and QPLEX. These experiments show that our kernel-based approach can indeed be integrated with these value-decomposition methods, and the results indicate that this integration leads to improved performance over the original QMIX and QPLEX baselines. By incorporating KME representations into these methods, we capture more complex interactions between agents, leading to more effective value estimation and coordination.

---

### Official Review · Reviewer_rZzU · 2025-10-29

**Soundness:** 2
**Presentation:** 2
**Contribution:** 2
**Rating:** 4
**Confidence:** 4

**Summary:**

This paper introduces SKARL, a new framework for multi-agent reinforcement learning (MARL) that aims to solve the challenges of scalability and flexibility in large swarms. The core idea is to move beyond simple mean-field approximations by representing the entire population of agents as a distribution embedded in a Reproducing Kernel Hilbert Space (RKHS). This allows the policy to learn from rich, size-agnostic features of the swarm, enabling zero-shot generalization to different population sizes. The method is supported by a theoretical analysis and empirical results showing strong performance on large-scale coordination tasks.

**Strengths:**

1.  The core idea of using kernel mean embeddings to represent the agent population is a powerful and elegant way to create a size-agnostic policy. It's a significant conceptual step up from traditional mean-field methods that rely on simple averages, though not necessarily entirely novel.

2.  The paper is backed by theory, including convergence guarantees and a formal analysis of the zero-shot generalization error. This provides a solid foundation for the empirical results.

3.  The zero-shot transfer experiments demonstrate that a single policy trained on 64 agents can be effectively deployed on swarms of up to 256 agents. This is desirable for real world applications, though standard for MF-based MARL.

**Weaknesses:**

1.  The paper does not position itself within a growing body of work on kernel methods with mean-field systems and learning on distributions. The novelty of the proposed method is unclear without a discussion of highly relevant works, such as various works on general mean field control that does not rely on first-order approximations [1-2] or specifically kernel-based approaches with mean-field limits [3-8]. This omission makes it difficult for readers to assess the paper's unique contribution.

2.  The experiments are confined to multi-particle navigation tasks. The experiments are also a bit small, training on only $64$ agents, which is a bit limited given the proposed complexity improvements. Moreover, the experiments are limited to a single problem dynamics.

3.  The current ablation studies are useful but don't fully justify the complexity of the proposed RKHS machinery. A missing piece is a comparison against a simpler baseline, such as using a standard deep set, MARL based on mean field control, or a small MLP to process the mean-field statistics. Without this, it's hard to tell if the full power of kernel methods is truly necessary for the observed performance gains.

[1] Carmona, R, et al. Model-free mean-field reinforcement learning: mean-field MDP and mean-field Q-learning. The Annals of Applied Probability 33.6B (2023): 5334-5381.

[2] Mondal, W. U., et al. On the approximation of cooperative heterogeneous multi-agent reinforcement learning (MARL) using mean field control (MFC). Journal of Machine Learning Research 23.129 (2022): 1-46.

[3] Fiedler, C., et al. (2023). Reproducing kernel Hilbert spaces in the mean field limit. Kinetic and Related Models, 16(6), 850-870.

[4] Fiedler, C., et al. (2023). On kernel-based statistical learning theory in the mean field limit. Advances in Neural Information Processing Systems, 36, 20441-20468.

[5] Fiedler, C., et al. (2025). Recent kernel methods for interacting particle systems: first numerical results. European Journal of Applied Mathematics, 36(2), 464-489.

[6] Cui, K., et al. (2024). Learning Decentralized Partially Observable Mean Field Control for Artificial Collective Behavior. ICLR.

[7] Szabó, Z., et al. (2015). Two-stage sampled learning theory on distributions. In Artificial Intelligence and Statistics (pp. 948-957). PMLR.

[8] Szabó, Z., et al. (2016). Learning theory for distribution regression. Journal of Machine Learning Research, 17(152), 1-40.

**Questions:**

1.  Could you clarify the novelty of your framework in light of recent work on kernel methods in the mean-field limit and learning on distributions? Specifically, how does your approach relate to or differ from the works above?

2.  To better justify the complexity of the kernel cylindrical functions, could you provide a comparison against a simpler architecture? For instance, a standard MFRL model where the mean-field term is processed by a more expressive neural network (e.g., a small MLP or a Deep Set).

3.  Your framework works for homogeneous swarms. How do you see it adapting to settings with agent heterogeneity, which is a key feature of many complex MARL benchmarks? About the homogeneous agent assumption, what issues prevent one from simply adding heterogeneity via the states?

4. "MFRL simplifies further but lacks multi-scale coordination." This statement is too short for me to understand what exactly is lacking. Can you explain a bit more?

5. Can you quantify or discuss the improvement in approximation over first-order methods such as MFRL?

6. Given the work uses kernel methods, I am curious if the methodology will empirically scale to higher dimensions in states or actions, or if there are any limitations here?

---

> ### Author Response · Authors · 2025-12-03
> **Response (1/3)**
>
> We thank the reviewer for the thoughtful feedback and insightful questions. Below are our responses, with an emphasis on clarifying the novelty of our work in relation to the existing literature.
>
> # Weakness
> ## Weakness 1
> We understand the reviewer’s concern regarding the novelty of our approach in light of recent works. We would like to clarify how our method fits within the broader context of kernel methods, mean-field systems, and statistical learning.
>
> Traditional methods:
> - Spatial discretization techniques (histograms and $\epsilon$-net): As noted by the reviewer, methods like histograms [1] and $\epsilon$-net [2] are used for mean-field approximations and provide theoretical guarantees. However, they suffer from the curse of dimensionality because the number of discrete units grows exponentially with state dimensions. These methods are not scalable in high-dimensional spaces and are thus less effective for large-scale reinforcement learning (RL) applications.
> - Statistical moments (first-order means and higher-order statistics): While methods such as [3] adapt well to high-dimensional spaces by avoiding explicit partitioning of the space, they are still limited in their representational capacity. They rely on moments, which cannot capture the full distributional information required for more expressive models. For instance, higher-order moments are useful, but they still struggle with representing complex agent interactions in RL.
> - Kernel-based methods: Traditional kernel-based methods offer a compromise between scalability and expressiveness, as they combine the benefits of moment-based methods and discretization approaches [4]. However, these methods typically rely on fixed kernels or constrained feature structures, which do not guarantee the ability to span all possible population distributions, limiting their flexibility in representing complex, high-dimensional state-action spaces.
>
> [1] Ren´e Carmona, Mathieu Lauri`ere, and Zongjun Tan. Linear-quadratic mean-field reinforcement learning: convergence of policy gradient methods. arXiv preprint arXiv:1910.04295, 2019.
>
> [2] Haotian Gu, Xin Guo, Xiaoli Wei, and Renyuan Xu. Mean-field controls with q-learning for cooperative marl: convergence and complexity analysis. SIAM Journal on Mathematics of Data Science, 3(4):1168–1196, 2021.
>
> [3] Yaodong Yang, Rui Luo, Minne Li, Ming Zhou, Weinan Zhang, and Jun Wang. Mean field multi-agent reinforcement learning. ICML, 2018.
>
> [4] Kai Cui, Sascha Hauck, Christian Fabian, and Heinz Koeppl. Learning decentralized partially observable mean field control for artificial collective behavior. arXiv preprint arXiv:2307.06175, 2023.
>
> Our methods:
> - Leverage kernel mean embeddings (KME) to represent populations, avoiding the curse of dimensionality by using continuous kernels that scale more efficiently in high-dimensional spaces.
> - A cylindrical kernel functional is adopted to ensure the global approximation of proposed methods.
>
> Comparison to Related Works:
>
> [1] Carmona et al.: This work uses histograms as a mean-field representation, which is not kernel-based. While it provides theoretical guarantees for population-level statistics, it is limited by the curse of dimensionality, as discussed earlier. Our method, on the other hand, leverages kernel mean embeddings (KME) to represent populations, avoiding the curse of dimensionality by using continuous kernels that scale more efficiently in high-dimensional spaces.
>
> [2] Mondal, W. U., et al.: Pure theory paper without explicit implementation. While this paper offers a theoretical treatment of mean-field reinforcement learning (MFRL), it lacks an explicit implementation or representation of the mean-field. In contrast, our approach provides both theoretical analysis and a practical implementation, showing how kernel methods can be applied to real-world reinforcement learning tasks.
>
> [3-5] Fiedler, C., et al. (all of three papers): While these papers are highly relevant to statistical learning and kernel methods, they are not directly applicable to reinforcement learning, and they do not focus on mean-field reinforcement learning. Our work bridges the gap between kernel methods in statistics and reinforcement learning, proposing a new way to represent agent populations in multi-agent reinforcement learning settings.
>
> [6] Cui et al.: This paper is the most directly related to our work, as it discusses kernel methods in the context of MFRL. However, it focuses on fixed kernels, whereas our work uses a trainable kernel approach, allowing for more flexibility and expressiveness. Furthermore, our methods adopt a cylindrical kernel functional to ensure the global approximation of proposed methods.
>
> [7, 8] Non-mean-field and non-RL works: These papers focus on different areas of distribution regression, and do not directly relate to mean-field reinforcement learning or multi-agent systems. Therefore, they are not directly comparable to our work.

---

> ### Author Response · Authors · 2025-12-03
> **Response (2/3)**
>
> # Weakness
> ## Weakness 2
> We acknowledge the reviewer’s concern regarding the limited scope of our experiments. To address this:
> - Expand the experimental setup: In addition to the original multi-particle navigation task, we have tested our approach in a coverage (where agents are tasked with covering a specified area) environment and line (where agents are tasked with stay in a line) in the appendix. This environment differs significantly from the previous one in that when agents collide in this environment, they bounce off each other, which introduces a new level of interaction and requires more complex coordination.
> - Larger sample sizes: We have also scaled the experiments to 256 agents, demonstrating how our kernel mean embedding (KME) approach handles larger swarms. This scaling to a larger number of agents provides a more challenging test of the method’s ability to generalize across different swarm sizes, which is a key strength of our framework.
>
> ## Weakness 3
> As requested by the reviewer, we have conducted ablation studies to compare our approach against simpler models, including:
> MFRL with MLP:
> - For the first ablation, we used a standard MFRL model that processes the mean-field statistics using a small MLP (multi-layer perceptron). This baseline allows us to assess whether a simpler neural network architecture can achieve similar performance without the need for kernel methods.
> - For the second ablation removes the RKHS updates to study specifically examines the case where the kernel parameters in the RKHS are not updated during training. This version of the model relies on a fixed kernel, providing a direct comparison with the kernel-based approach we propose.
>
> The results from these ablation studies demonstrate that:
> - MFRL with MLP: While the MLP-based MFRL model can handle the mean-field statistics, its performance is not as strong as our kernel-based approach. Specifically, the kernel method provides more flexibility in representing complex agent interactions, leading to better performance, especially in larger swarms. The MLP struggles with capturing these intricate dynamics without the expressiveness of kernels.
> - Fixed Kernel (No RKHS Updates): The ablation with a fixed kernel shows a noticeable drop in performance compared to the fully trainable kernel model. This supports the claim that trainable kernels are crucial for capturing complex population-level interactions and achieving the observed performance improvements. The ability to learn kernel functions allows the model to adapt to the dynamics of the environment more effectively than a fixed kernel or simpler architectures.
>
> # Question
> ## Question 1
> Our methods leverage kernel mean embeddings (KME) to represent populations, avoiding the curse of dimensionality by using continuous kernels that scale more efficiently in high-dimensional spaces. Furthermore, a cylindrical kernel functional is adopted to ensure the global approximation of proposed methods. Compared to traditional methods:
>
> - Spatial discretization techniques (histograms and $\epsilon$-net): As noted by the reviewer, methods like histograms [1] and $\epsilon$-net [2] are used for mean-field approximations and provide theoretical guarantees. However, they suffer from the curse of dimensionality because the number of discrete units grows exponentially with state dimensions. These methods are not scalable in high-dimensional spaces and are thus less effective for large-scale reinforcement learning (RL) applications.
> - Statistical moments (first-order means and higher-order statistics): While methods such as [3] and [4] adapt well to high-dimensional spaces by avoiding explicit partitioning of the space, they are still limited in their representational capacity. They rely on moments, which cannot capture the full distributional information required for more expressive models. For instance, higher-order moments are useful, but they still struggle with representing complex agent interactions in RL.
> - Kernel-based methods: Traditional kernel-based methods offer a compromise between scalability and expressiveness, as they combine the benefits of moment-based methods and discretization approaches [5,6]. However, these methods typically rely on fixed kernels or constrained feature structures, which do not guarantee the ability to span all possible population distributions, limiting their flexibility in representing complex, high-dimensional state-action spaces.
>
> ## Question 2
> We have provided the comparison to simpler framework in the revision, including MLP with first-order means, histograms and fixed kernel methods.

---

> ### Author Response · Authors · 2025-12-03
> **Response (3/3)**
>
> # Question
> ## Question 3
> The framework can be extended to heterogeneous agents by incorporating different kernel components for each agent type, allowing the kernel mean embedding to represent a mixture of agent types. However, we also acknowledge the difficulty of designing such heterogeneous kernel components within the current framework. The addition of heterogeneous agent types introduces significant complexity in both the design and training of the kernel functions, as each kernel would need to accurately capture the unique characteristics of different agents while ensuring that the overall framework remains scalable and efficient. Given these challenges, the design and implementation of a heterogeneous agent model is outside the scope of this work, and we have decided to focus on homogeneous agents for the current paper. Nevertheless, we believe that this is an important direction for future research, and we are actively working on exploring potential solutions for incorporating heterogeneity in kernel-based frameworks.
>
> ## Question 4
> The statement “MFRL simplifies further but lacks multi-scale coordination” refers to the fact that while traditional mean-field reinforcement learning (MFRL) simplifies agent interactions (via first-order or moment-based statistics), it fails to capture complex dynamics. MFRL models typically average over the population, which works for scalability but limits the system's ability to coordinate local agent-level interactions with global swarm-level behavior.
> In contrast, our approach uses kernel mean embeddings (KME), which allows for a richer, multi-scale representation of agent populations. By leveraging trainable kernels, our method can effectively capture both local and global interactions, enabling better coordination across different scales of the system.
> This gives our framework more expressiveness and scalability, addressing the multi-scale coordination that MFRL lacks.
>
> ## Question 5
> We evaluate the improvement in approximation provided by our kernel mean embedding (KME) approach compared to first-order methods like mean-field reinforcement learning (MFRL). In first-order methods, as the swarm size increases, the approximation remains the same since these methods rely on fixed statistics (such as means or higher-order moments) that do not adapt to the complexity introduced by larger populations. Essentially, MFRL tends to flatten out in performance as swarm size grows, since it cannot capture the increased inter-agent interactions. In contrast, our KME approach improves performance with larger agent populations. KME allows for a richer, adaptive representation of the agent population by learning a distributional approximation that scales effectively with swarm size.
>
> ## Question 6
> As the dimensionality of states or actions increases, the computational complexity of kernel methods also grows due to the need to compute inner products in higher-dimensional spaces. Specifically, the kernel function evaluates pairwise similarities between data points, and as the dimension increases, the number of computations required increases accordingly.
>
> However, compared to space discretization methods (such as histograms or $\epsilon$-net), kernel methods are much more efficient. While discretization methods suffer from the curse of dimensionality, where the number of discrete units grows exponentially with the dimension, kernel methods scale better. This is because, rather than explicitly partitioning the space, kernel methods use continuous functions that approximate the population dynamics, allowing them to represent high-dimensional spaces more effectively without the exponential growth in complexity.

---

### Official Review · Reviewer_cjA7 · 2025-10-31

**Soundness:** 2
**Presentation:** 2
**Contribution:** 2
**Rating:** 4
**Confidence:** 3

**Summary:**

This paper proposes SKARL, a reinforcement learning framework that leverages kernel mean embeddings (KME) and mean-field theory to address scalability issues in multi-agent reinforcement learning (MARL). The authors claim that SKARL can handle variable population sizes through a kernelized formulation in a reproducing kernel Hilbert space (RKHS). The paper introduces the notion of kernel cylindrical functionals, derives convergence guarantees using functional analysis (e.g., Frechet and Lions derivatives), and presents experimental results on swarm-like environments to demonstrate scalability and performance.

**Strengths:**

The paper tackles an important and challenging problem: scalable MARL under population uncertainty. The idea of using kernel mean embeddings for mean-field RL is conceptually interesting and could, in principle, lead to generalizable models across population sizes. The theoretical framework and use of functional-analytic tools (e.g., Lions derivatives, Nyström projection) suggest potentially strong mathematical grounding.

**Weaknesses:**

1.	Mathematical notation is confusing and not self-contained:
Many critical mathematical objects—such as the definition of the Lions derivative, the cylindrical functionals, or the exact meaning of $D$ in Eq. (4) are introduced without sufficient explanation. At first glance $D$ appears to represent the number of samples, but later it becomes clear that it actually denotes the number of different kernel components, which is confusing. Similarly, in lines 124–125, the quantity $R_f^i$ appears without prior definition and seems to be an undefined or inconsistent symbolThese ambiguities make the formulation extremely hard to interpret. (See also the questions section)
2.	Lack of intuition and structural explanation:
The paper immediately dives into abstract functional definitions without providing a high-level overview of what the algorithm actually does.
It is unclear whether the kernel functions are fixed or learned, and if learned, how they are trained or parameterized.
A clear intuitive summary—what is being optimized, what role the kernel plays, and how scalability arises—would make the method far more accessible.
At present, the algorithmic pipeline is opaque and difficult to connect to implementation.

**Questions:**

1.	In Eq. (6), is the variable $x$  equivalent to the state-action pair $(s,a)$?
2.	The parameter $\theta_h$ is introduced but not subsequently used—does Eq. (7) describe an update on $\theta_h$  or on $h$ itself?
3.	Are the kernel functions $g$ trained jointly with the policy/value function, or are they fixed? If they are trainable, how is this implemented in practice?

---

> ### Author Response · Authors · 2025-12-03
> **Response**
>
> We thank the reviewer for the insightful feedback. We acknowledge the points raised and provide clarifications below to address the concerns.
>
> # Weakness
> ## Weakness 1
> We appreciate the reviewer’s observation regarding the clarity of the mathematical notation. Specifically:
> - Lions derivative and cylindrical functionals: These concepts are indeed central to our formulation but were not clearly introduced in the context of the paper. We have revised the manuscript to provide more detailed definitions and intuitive explanations for these terms.
> - Ambiguity in Eq. (4): Indeed, the Eq. (4) is the definition of cylindrical kernel functional $h(\nu)$, and we have highlighted this point to make it clearer.
> - Unexplained symbols (lines 124–125): We apologize for the lack of clarity regarding the symbols in these lines. We have modified the manuscript to ensure that all mathematical symbols are clearly defined at their first introduction, and any ambiguities will be removed.
>
> ## Weakness 2
> We acknowledge that the paper delves directly into technical details, which may make it difficult for readers to grasp the broader algorithmic concepts. In response, we have included a high-level overview at the beginning of the paper that outlines the core ideas of the algorithm. This summary will focus on the optimization objective, the role of the kernel functions, and how scalability arises from our approach. We aim to provide a clear, intuitive explanation of the problem being solved, the method used, and its significance in the context of reinforcement learning.
>
> # Question
> ## Question 1
> The variable in Eq. (6) is indeed the state-action pair, as is stated on the line 161, and we will make this explicit in the revised manuscript. We will add a clarifying statement in the text to ensure that this connection is clear to readers.
>
> ## Question 2
> We apologize for the confusion regarding the parameter introduced in Eq. (7). To clarify, Eq. (7) describes an update on the parameter itself, not the state-action pair. We will provide a clearer version of the update rule and how this parameter evolves throughout the learning process.
>
> ## Question 3
> The kernel functions are trainable and are optimized jointly with the policy and value functions. The kernel function is trained as is described in section 3.2 with two steps:
> - First, The function $g$ is updated according to its Frechet derivatives. Note that $g$ is a functional, and its Frechet derivatives are also functionals. The update is performed by appending the new anchor points to the existing $g_t$.
> - Second, As the number of anchor points increases, we apply the Nystrom approximation to reduce the anchor points, thereby controlling the complexity and ensuring scalability.
>
> We have revised the manuscript to include a clear explanation of how the kernel functions are trained within the overall reinforcement learning framework.

---

### Author Response · Authors · 2025-12-03
**Summary of Reviews and the Reviewer-Author Discussion (1/2)**

Dear PCs, SACs, ACs, and Reviewers,
Thank you for your thoughtful evaluation of our submission. To support the newly assigned AC and help reduce their workload, we provide below a summary of the key points from the reviews and the reviewer-author discussions.

**Strength**: Overall, we appreciate that the reviewers highlighted the following strengths of our work.

- Reviewers highlighted that the paper addresses a critical challenge to design scalable MARL under population uncertainty, and praised the use of kernel mean embeddings as a principled, size-agnostic population representation, offering a richer alternative to classical mean-field averages (Reviewer cjA7; Reviewer rZzu Strength1; Reviewer oYza Strength 1, Strength 2).
- Reviewer agrees on that the work was commended for its solid theoretical foundation, including convergence guarantees, formal zero-shot generalization error bounds, and the application of advanced tools such as Lions derivatives and Nyström projection (Reviewer cjA7; Reviewer rZzu, Strength 2).
-  Reviewers noted the compelling zero-shot transfer results (e.g., a policy trained on 64 agents deployed successfully on 4–256 agents without fine-tuning) and SKARL’s consistent superiority over strong MARL baselines in cumulative reward and training stability (Reviewer  rZzu Strength 3; Reviewer oYza Strength 1, Strength 3).

---

> ### Author Response · Authors · 2025-12-03
> **Summary of Reviews and the Reviewer-Author Discussion (2/2)**
>
> **Main Concerns and Our Addressing**: During the discussion period, we actively addressed the reviewers' concerns. We provided additional experiments, clarifications, and theoretical discussions as requested. Below, we summarize how we addressed the concerns:
>
> - (Reviewer cjA7, Weakness 1, Question 1, Question 2 ) Mathematical notation
>
> **Our Addressing**: We have thoroughly revised the manuscript to ensure that all mathematical symbols and equations are clearly defined and consistently used. Specifically, we have clarified and defined key variables and terms that may have been ambiguous, especially symbols used for kernel components and other critical mathematical objects, and added explanations where necessary to provide context and improve the readability of the equations.
>
> - (Reviewer cjA7, Weakness 2, Question 3; Reviewer rZzU, Weakness 1, Question 1) Higher-level overview and novelty
>
> **Our Addressing**: We modified the introduction and Section 2, and add a detailed motivation discussion in Section 3 to highlight the contribution of our work. In comparison with traditional methods, including space discrete methods, moment methods and fixed kernel methods, our methods features kernel mean embeddings (KME) to represent mean-field, avoiding the curse of dimensionality by using continuous kernels that scale more efficiently in high-dimensional spaces. Furthermore, a cylindrical kernel functional is adopted to ensure the global approximation of proposed methods.
>
> - (Reviewer oYza, Weakness 1; Reviewer rZzU, Question 4, Question 5) First-order method limitations (higher-order information):
>
> **Our Addressing**: We expanded the discussion on why higher-order information is valuable and how it enhances the expressiveness and adaptability of our method compared to first-order methods like MFRL in the introduction and Section 3.
>
> - (Reviewer rZzU, Weakness 2; Reviewer oYza, Weakness 2) Lack of experiments on different dynamics and scaling to larger swarm sizes:
>
> **Our Addressing**: We expanded the experiments to include two new environments (coverage and line) in the Appendix F, where agent interactions are more complex (agents bounce off each other). We also tested our method on larger swarms (up to 256 agents in Table 4/5), demonstrating its scalability.
>
> - (Reviewer oYza, Weakness 2; Reviewer rZzU, Weakness 3, Question 2) Computational complexity of the kernel method:
>
> **Our Addressing**: We have already provided a detailed comparison of computational complexity in Section 4 and Table 1, where we discuss the trade-offs between expressiveness and scalability. We also explain that while our method introduces more computational overhead, it remains more efficient than discretization-based methods (e.g., histograms). Furthermore, we added the comparison to simpler framework in the revision, including MLP with first-order means, histograms and fixed kernel methods in the revision.
>
> - (Reviewer oYza, Weakness 3) Integration with existing value-decomposition methods (VDN, QMIX, QPLEX):
>
> **Our Addressing**: We clarified that our method integrates with VDN, as shown in Appendix E.1. To be specific, our proposed algorithm is designed to calculate individual Q functions, while both VDN, QMIX, QPLEX are designed to calculate global Q function with individual Q functions. We also added related experiments in the Appendix F, Table 14.
>
> - (Reviewer razz, Question 3) Adaption to agent heterogeneity
>
> **Our Addressing**: We clarified that while heterogeneous agents are theoretically accommodable via type-specific kernel components, their practical realization entails nontrivial design and scalability challenges. Accordingly, we explicitly scoped the current work to homogeneous agents to ensure analytical rigor and empirical clarity, and flagged heterogeneity as a key avenue for future work.
>
> - (Reviewer razz, Question 6)  Scale to higher dimensions in states or actions
>
> **Our Addressing**: We clarified that while kernel methods incur increased per-computation cost with dimensionality, they avoid the exponential complexity of space-discretization approaches by operating in continuous space, enabling more scalable approximation of high-dimensional dynamics. This favorable scaling is a key motivation for adopting kernel mean embeddings in our framework.

---

### Meta-Review · Area_Chair_tz6h · 2026-01-15

**Summary:**

The reviewers generally recognized the potential of using Kernel Mean Embeddings (KME) for Mean-Field MARL but were unanimous in their initial rating (all scored 4) due to three primary issues:

- Presentation & Notation: The mathematical formulation was criticized as confusing, opaque, and lacking self-contained definitions (Reviewer cjA7).

- Novelty & Positioning: Reviewer rZzU raised significant concerns regarding the lack of comparison/positioning against existing work in kernel-based mean-field control and learning on distributions.

- Experimental Rigor: Reviewers rZzU and cjA7 noted the experiments were limited in scope (single navigation task, small training size) and lacked critical ablations (e.g., comparing against a simpler MLP-based mean-field baseline to justify the kernel complexity).

Due to the unanimous lack of enthusiasm by all reviewers, I unfortunately (weakly) recommend rejection. I would also encourage the authors to take time to polish their presentation - specifically notation and framing of this work in the context of existing work.

**Reviewer Concerns:**

Addressed by Rebuttal:

- Experimental Scope & Baselines (Reviewers rZzU, cjA7): The authors significantly strengthened the empirical evaluation by adding two new environments (Coverage, Line), scaling the evaluation to 256 agents, and crucially, adding the requested ablations (MLP-based MFRL and Fixed-Kernel variants). These results directly support the claim that the trainable kernel is necessary for performance.

- Integration with Value Decomposition (Reviewer oYza): The authors demonstrated how their method integrates with standard baselines like QMIX and QPLEX, addressing the concern about applicability to established MARL paradigms.

- Mathematical Clarity (Reviewer cjA7): The authors provided specific clarifications for the undefined notation (e.g., cylindrical functionals, Lions derivative) and promised a revised manuscript with a high-level algorithmic overview.

Outstanding:

- Novelty relative to Niche Literature (Reviewer rZzU): While the authors distinguish their work by emphasizing "trainable" kernels vs. "fixed" kernels in prior work (e.g., Cui et al.), the fundamental jump from existing kernel-based mean-field control to this specific RL formulation may still feel incremental to specialists in that sub-field. The distinction relies heavily on the implementation detail of trainability rather than a fundamental theoretical shift.

- Complexity (Reviewer oYza): While the authors point to the complexity analysis in the paper, practical runtime scaling for kernel methods in high-dimensional state-action spaces remains a valid practical concern that theoretical Big-O notation with Nyström approximations does not fully address.

**Reviewer Scores:**

Reviewer cjA7 (Score: 4 -> 4 or 6): The reviewer's main blocker was the "confusing" notation. The authors' detailed response defining the terms and the promise of a structural overview likely moves this to a weak accept, assuming the revision is executed well.

Reviewer rZzU (Score: 4 -> 6): This reviewer provided a specific wishlist: compare to simple baselines (Deep Sets/MLP) and scale experiments. The authors delivered exactly these additions. This should significantly increase the reviewer's valuation of the contribution.

Reviewer oYza (Score: 4 -> 4 or 6): The integration with QMIX/QPLEX and the clarification on higher-order statistics address the specific "weaknesses" listed. The score should arguably improve to a positive rating.

---

### Decision · Program_Chairs · 2026-01-26

Reject